



# A fine-resolution soil moisture dataset for China in 2002-2018

Xiangjin Meng[1,2,★], Kebiao Mao[1,3,★], Fei Meng[4], Jiancheng Shi[5,6], Jiangyuan Zeng[6], Xinyi Shen[7], Yaokui Cui[8], Lingmei Jiang[6], and Zhonghua Guo[1]

[1] School of Physics and Electronic-Engineering, Ningxia University, Yinchuan 750021, China.
[2] School of Earth Sciences and Engineering, Hohai University, Nanjing 211100, China.
[3] Institute of Agricultural Resources and Regional Planning, Chinese Academy of Agricultural Sciences, Beijing, 100081, China
[4] School of Surveying and Geo-Informatics, Shandong Jianzhu University, Jinan, 250100, China.
[5] National Space Science Center, Chinese Academy of Sciences, Beijing, 100190, China.
[6] State Key Laboratory of Remote Sensing Science, Jointly Sponsored by the Aerospace Information Research Institute of Chinese
Academy of Sciences and Beijing Normal University, Beijing, 100101, China.
[7] Civil and Environmental Engineering, University of Connecticut, Storrs, CT 06269, USA.
[8] School of Earth and Space Sciences, Peking University, Beijing, China, 100871.
*Correspondence to:* Kebiao Mao (maokebiao@caas.cn)
★ These authors contributed equally to this work.

**Abstract:** Soil moisture is an important parameter required for agricultural drought monitoring and climate change models. Passive microwave remote sensing technology has become an important means to quickly obtain soil moisture over large areas, but the coarse spatial resolution of microwave data imposes great limitations on the application of these data. We provide a unique soil moisture dataset (0.05°, monthly) for China from 2002-2018 based on reconstruction model-based downscaling techniques using soil moisture data from different passive microwave products (including the

AMSR-E/2 Level 3 products and the SMOS-INRA-CESBIO (SMOS-IC) products) calibrated with a consistent model in combination with ground observation data. This new fine-resolution soil moisture dataset with a high spatial resolution overcomes the multisource data time matching problem between optical and microwave data sources and eliminates the difference between the different sensor observation errors. The validation analysis indicates that the accuracy of the new dataset is satisfactory (bias: -0.024, -0.030 and -0.016 m³/m³, unbiased root mean square error (ubRMSE): 0.051, 0.048

and 0.042, correlation coefficient (*R*): 0.82, 0.88, and 0.90 on monthly, seasonal and annual scales, respectively). The new dataset was used to analyze the spatiotemporal patterns of soil water content across China from 2002 to 2018. In the past 17 years, China's soil moisture has shown cyclical fluctuations and a downward trend (slope=-0.167, *R*=0.750) and can be summarized as wet in the south and dry in the north, with increases in the west and decreases in the east. The reconstructed dataset can be widely used to significantly improve hydrologic and drought monitoring and can serve as an

important input for ecological and other geophysical models. The data are published in the Zenodo at http://doi.org/10.5281/zenodo.4049958 (Meng et al., 2020)

**Keywords:** Downscaling, Soil moisture, Passive microwave, Spatial weighted decomposition (SWD) model, China, TVDI



## 1 Introduction

Soil moisture (SM), which is one of the key variables in the water cycle and atmospheric energy budget (Entekhabi et al.,

1999; Taylor et al., 2011; Shi et al., 2012; Guillod et al., 2015), has been widely used for flood forecasts (Bindlish, et al., 2009), drought detection (Mao, et al., 2010), crop yield estimation (Chen, et al., 2011), weather prediction and hydrological modeling (Liu, et al., 2017). Therefore, accurately monitoring and assessing the dynamics of the spatiotemporal distribution of SM are crucial for understanding the hydrological, ecological, and biogeochemical processes associated with global and regional climate systems (Mao et al., 2008b; Seneviratne et al., 2010; Han et al.,

2012; Wang et al., 2016). The most direct way to obtain SM is primarily from in situ measurements with SM measuring instruments at ground meteorological stations (Franz et al., 2012). SM networks based on ground stations have made great contributions to establishing long-term SM datasets (Srivastava 2016). The in situ SM observations from these networks have also been unified into a common database (Dorigo et al., 2011). However, the accurate measurements of SM are limited by the number of field sites around the world, and measuring SM at a single location does not necessarily

represent the condition of the entire region due to the large spatial heterogeneity of SM (Crow et al., 2002; Njoku et al., 2003). With the development of remote sensing technology, satellite-based SM measurements have become increasingly available, such as microwave observations from active and passive sensors, which is one of the most effective means for the retrieval of soil moisture (Loew, et al., 2011; Petropoulos, et al., 2015; Srivastava, et al., 2017). Active microwave remote sensing technology measures the energy reflected from the surface of the land after actively transmitting

microwave energy pulses, while passive microwave sensors measure the self-emitted energy from the land surface (Schmugge et al., 1974; Moran, et al., 2004; Shi et al., 2006; Shen et al., 2013; Bhagat et al., 2014). Both active and passive microwave remote instruments, particularly at low frequencies, have been used to provide global coverage surface soil moisture datasets (Njoku, 2003; Albergel, et al., 2013). The European Space Agency's Water Cycle Multi Mission Observation Strategy (ESA WACMOS) Support to Science Element (STSE) program has developed the first long-term

SM data record from passive and active microwave data. In 2012, the ESA's Climate Change Initiative (CCI) program SM datasets were first publicized on the ESA CCI web portal (Su et al., 2010). This CCI product was generated by merging different microwave sensor observations and attempting to produce a complete and consistent long-term time series of SM datasets (Dorigo et al., 2017; Gruber et al. 2019). Since then, it has been constantly updated, and the latest release (v05.2) provides global SM data up to 31-12-2019. These long-term availability of SM products has been validated

against extensive model simulations or in situ measurements (Albergel et al., 2012; Loew et al., 2013; Zeng et al., 2015; Dorigo et al., 2017) and are widely used for a range of soil moisture-related studies, such as climate model evaluation and drought monitoring.

The mentioned SM datasets have a coarse spatial resolution (e.g., 25 km), whereas a high-resolution SM product that can be directly used in hydrological process models (e.g., surface evapotranspiration models, land surface process models,



and agricultural drought models) in regional-scale or local-scale studies is needed to provide additional monitoring details,

unless a fine-resolution land surface SM product (e.g., from 1 to 10 km) is available. Downscaled SM data can help solve

the problem of coarse spatial resolution and are required for many regional agricultural and hydrological applications

(Sandholt et al., 2002; Peng et al., 2015; Mohanty et al. 2017). To improve the spatial resolution of passive microwave

SM data, various methods have been proposed to downscale SM. The principle of this is to construct a statistical

correlation or physical model relationship between coarse-resolution SM data and fine-resolution auxiliary variables to

achieve scale conversion (Jin et al., 2017). Some of these studies have tried to explore the relationship between optical

remote sensing products with a relatively fine spatial resolution and microwave remote sensing SM data with a coarse

spatial resolution (Maltese et al. 2015, Wang et al. 2016). Due to the difference in coverage between vegetation and bare

soil, the sensitivity of land surface temperature (LST) to SM changes varies, and the shapes of plotted LST and

normalized difference vegetation index (NDVI) data are usually presented in a physical sense as trapezoidal or triangular

feature space. Based on the LST/NDVI feature space, the temperature vegetation dryness index (TVDI) has been

developed to estimate the SM. Meanwhile, the TVDI has been widely used for the downscaling of microwave SM and

drought monitoring over different regions (Chauhan et al. 2003;). However, a significant problem in the downscaling

process is the time matching of different sensors, that is, the temporal gap between the coarse-resolution microwave

product and the fine-resolution optical product. This interval can cause a lager deviation between downscaled products

and original product, such as differences in daytime and nighttime surface temperatures, humidity, evapotranspiration,

water and heat. This requires obtaining SM and auxiliary optical/infrared (IR) data with relatively consistent time points.

It is generally difficult to aggregated data from long-sequence multisource sensors while taking into account most of the

sensors. The Aqua satellite, which is equipped with both the Advanced Microwave Scanning Radiometer - Earth

Observing System (AMSR-E) and the Moderate Resolution Imaging Spectroradiometer (MODIS) sensors, can

simultaneously provide coarse-resolution passive microwave SM and LST/NDVI data, providing a guarantee that the data

were acquired at the same time. However, AMSR-E data alone are insufficient (the instrument stopped working in

October 2011), and thus we used its successor AMSR2 to continue the data series. The missing data between AMSR-E

and AMSR2 (from November 2011 to June 2012) is supplemented by SMOS-IC data, which has been verified to have

higher accuracy by using in situ measurements across the globe (Al-Yaari et al., 2019; Ma et al., 2019). Many methods

have been proposed to handle these systematic differences among SM products from different microwave sensors

(Zwieback, et al., 2016). Recent studies have exploited the utility of rescaling SM product methods (Brocca et al., 2013;

Zeng et al., 2020). Linear regression rescaling of SM has proven to be a simple and effective method, and a review of

these rescaling methods has been published by Afshar et al. (2017).

In this study, all microwave SM data are based on AMSR-E Level 3 data, uniformly corrected to the same time and the

same depth of detection using a linear regression method. In addition, ground station data are incorporated, and a large

area of missing and invalid pixels is restored, so that the entire dataset is guaranteed to be complete in the Chinese region.





A spatial downscaling method, namely, the spatial weight decomposition (SWD) model, was utilized to decompose the inconsistencies in soil depth and time in the coarse spatial resolution SM products with the TVDI into SM data with a

0.05° spatial resolution. The dataset covers the period from 2002 to 2018 and is comprehensively compared with in situ SM datasets.

## 2 Study area

Most of the areas in China, which is located within the central and eastern parts of Asia and is situated along the western coast of the Pacific Ocean, are affected by the monsoon climate and have significant monsoon climate characteristics

(Feng et al., 2003). Drought disasters in China have constantly increased over the past few years, and drought has become one of the most serious types of natural disasters. Rapid increases in industrial, irrigation and domestic water use have resulted in dramatic increases in water resource consumption, which in turn have led to a significant increase in droughts in much of China, especially northern China (Zhao, et al., 2017). Thus, there is an urgent need to improve our knowledge about the spatial and temporal variability of SM in order to provide a basis for quantification and prediction (Liu, et al.,

2012). Hence, it is necessary to construct a set of high-precision and high spatial resolution SM datasets in China.

To study the spatial and temporal patterns of SM throughout the various regions of China, we divided China into six regions according to their climate conditions and topography: Northeast Monsoon Region (NEM), North China Monsoon Region (NCM), South China Monsoon Region (SCM), Southwest Humid Region (SWH), Northwest Arid Region (NWA), and Qinghai-Tibet Plateau Region (QTP) (Liang, et al., 2017). The Northeast Monsoon Region includes the areas to the

south of the Heilongjiang River, to the east of the Daxinganling Mountain range and to the north of the Ming Great Wall (117-135 °E, 38-53 °N). The North China Monsoon Region, which extends from the Inner Mongolia Plateau to the northern part of the Qinling-Huaihe River, east to the eastern part of the Yellow Sea and the Bohai Sea, and west to the eastern part of the Qinghai-Tibet Plateau, has typical temperate monsoon climate characteristics (103-125 °E, 33-42 °N). The South China Monsoon Region includes the monsoon region to the east of the Yunnan-Guizhou Plateau and to the

south of the Qinling Mountains-Huaihe River; this region has abundant rainfall and dense river networks and is characterized by a typical subtropical monsoon climate (20-33 °E, 105-123 °N). The Southwest Wet Region includes the Qinghai-Tibet Plateau and the Yunnan-Guizhou Plateau to the south of the Huaihe River and the Sichuan Basin; precipitation is abundant in southwestern China (21-34 °E, 97-104 °N). The Northwest Arid Region includes the Inner Mongolia Plateau to the east of the Greater Xing'an Mountains and the vast arid and semiarid regions of Northwest China

in the Tarim Basin to the north of the Qinghai-Tibet Plateau (73–126 °E, 37–55 °N). The Qinghai-Tibet Plateau Region includes the southern part of the Kunlun Mountains-Altun Mountains-Qilian Mountains, the area to the west of the Hengduan Mountains, and the entire Qinghai-Tibet Plateau to the north of the Himalayas (73–104 °E, 27–40 °N, Zhao and Chen 2011). The locations of the meteorological stations and six geographic-climatic regions in China are shown in Figure 1. For each region, we analyzed the current SM conditions and their changes over the past 17 years.

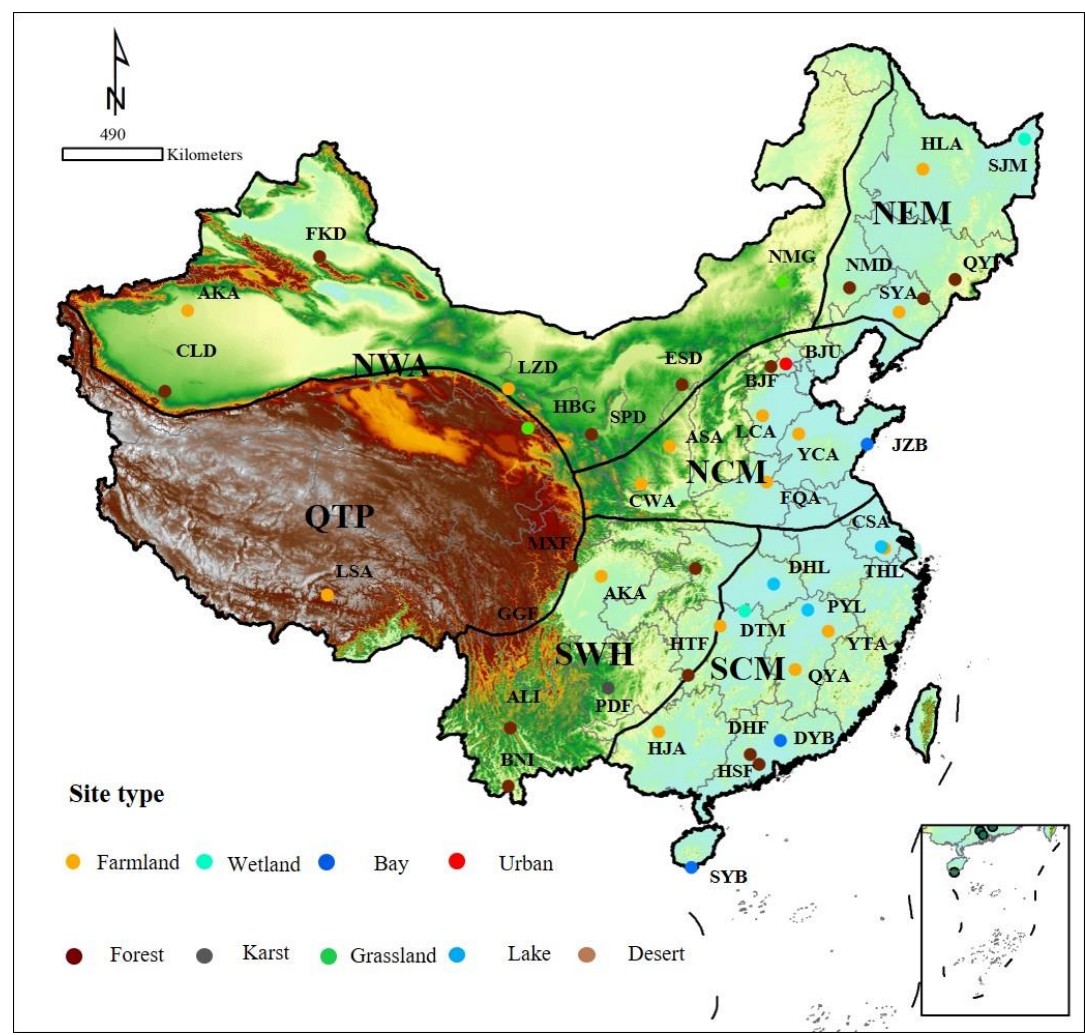


Figure 1: Overview of the study area, location of the meteorological stations and six geographic-climatic regions in China

Note: More detailed information on dense sites can be found in Table S1. Base map is derived from the Resource and Environment Science and
Data Center of Chinese Academy of Sciences (http://www.resdc.cn/)

## 3 Data and methodology

3.1. Satellite-derived SM Data

Since satellite sensors have a finite working life, to obtain a longer sequence of SM data sets, we need to use different

satellite sensors to generate SM products. The SM data are mainly derived from the AMSR-E/2 Level 3 and SMOS-IC,

which have moisture units of m³/m³ and spatial resolutions of 0.25°. Among these satellite sensors, AMSR-E is aboard the

Aqua satellite (effective service period from May 2002 to October 2011) with transit times of 13:30 and 01:30, and the

orbit is a sun-synchronous near-polar orbit with an orbital height of approximately 700 km (Kim, et al., 2012; Rüdiger et

al., 2009). AMSR-E has six wavelengths in the microwave spectrum (6.925, 10.65, 18.7, 23.8, 36.5, and 89 GHz). The

SM data utilized in the current research were obtained from the Japan Aerospace Exploration Agency (JAXA) AMSR-E

SM L3 product (Koike et al., 2004), and the time series ranges from July 2002 to September 2011. This product is based





on the JAXA algorithm, and posted on a 0.25°. First, a forward radiative transfer scheme is used to establish a brightness

temperature data set for a variety of frequencies and polarization-generated parameter values (soil and vegetation). Then,

the brightness temperature data set is used to create a lookup table (LUT). Finally, the SM and vegetation water content

are estimated by using the microwave polarization difference index (MPDI) at 10.65 GHz and the index of soil wetness

(ISW) at 36.5 GHz and 10.65 GHz horizontal channels (Koike et al. 1996, 2004). The JAXA algorithm assumes that the

optical depth of vegetation is linearly related to the vegetation water content and that the vegetation water content can be

determined by the NDVI. Based on the verification of the ground monitoring network, JAXA products provide acceptable

SM results (Zeng et al. 2015).

The AMSR-2 sensor is mounted on the Japanese Global Change Observation Mission – Water Satellite 1 (GCOM-W1) and

was launched in May 2012. As a follow-up to AMSR-E, AMSR-2 has a larger antenna reflector diameter, increasing from 1.6

m to 2.0 m. Moreover, AMSR-2 includes an extra C-band channel (with a frequency of 7.3 GHz) to mitigate radio frequency

interference (RFI). The transit times are still 13:30 and 01:30. The data were derived from the JAXA SM products, which

were released in real time, and the time span ranges from July 2012 to December 2018. As a continuation of the AMSR-E

product, the AMSR-2 L3 product also uses an LUT method to obtain SM retrievals, providing two products with spatial

resolutions of 0.1° and 0.25°. To better match the available data, this paper selects the 0.25° spatial resolution data. The

accuracy of the JAXA AMSR2 product was verified to have a root mean square error (RMSE) of less than 0.06 $m^3/m^3$ (for

a vegetation water content of ≤1.5 $kg/m^2$, Kim, et al. 2015).

The SMOS satellite was launched on November 2, 2009. This satellite travels along a sun-synchronous orbit with an

average altitude of 758 km and a dip of 98.44°. The transit times are approximately 06:00 (ascending) and 18:00

(descending) local time with a two- to three-day revisit frequency. The operating L-band (1.4 GHz), measured with a

Microwave Imaging Radiometer with Aperture Synthesis (MIRAS), is used to observe SM (Kerr et al., 2012; Lacava, et

al., 2012; González-Zamora et al., 2015). This study uses the SMOS-IC V105 SM product contributions from the Centre

Aval de Traitement des Données SMOS (CATDS), with a time series ranging from October 2011 to December 2018 and a

spatial resolution of 25km. SMOS-IC algorithm was designed by the Institut National de la Recherche Agronomique

(INRA) and Centre d'Etudes Spatiales de la BIOsphère (CESBIO) (Fernandez-Moran et al., 2017). The SMOS-IC

product was further quality filtered and redivided based on the previous SMOS Level 2 SM user data product (SMDUP2)

algorithm. That is, the values of the grid point data quality index (DQX) greater than 0.07, which were affected by RFI or

SM, were discarded; then, the DQX reverse-weighted average was used to group the SMDUP2 data on a 0.25° equal-area

grid and obtain SMOS-IC-grade products with a 25km spatial resolution. Al-Yaari et al. (2019) and Ma et al. (2019)

conducted comprehensive evaluation of the SMOS-IC SM product by using ground measurements worldwide. The results

showed that the SMOS-IC SM product agreed better with in situ measurements than other SMOS products (SMOS L2

and L3). The SMOS-IC scientific data were as independent as possible from auxiliary data, and the data are available at

https://www.catds.fr/Products/Available-products-from-CEC-SM/SMOS-IC. The data are provided on a daily time scale



to match the AMSR-E/2 L3 SM products at the same scale. The SMOS-IC SM data were aggregated to a monthly temporal resolution.

3.2 MODIS LST and NDVI Data

In a downscaling model, it is critical to establish the relationships between SM and other high-resolution surface variables. Im et al. (2016) utilized the relationships between SM and MODIS-derived products to improve the resolution of the AMSR-E SM product. Wang et al. (2016) downscaled SM data from a 0.25° resolution to a 0.05° resolution using a similar approach. Zhao et al. (2018) used the vegetation-thermal relationship to establish a microwave-optical/infrared downscaling model to optimize the spatial resolution of SMAP SM products to a very good level of precision. All these

land surface variables are available from the corresponding MODIS products. The MODIS sensor aboard the Aqua satellites passes over China at approximately 01:30 (descending) and 13:30 (ascending). MODIS has been widely used to monitor various environments, including land, oceans, and the lower atmosphere, due to its high temporal resolution and good data quality. In this study, two MODIS products were used, namely, the MODIS/AQUA monthly LST (MYD11C3) and NDVI (MYD1C2) products, which have 0.05° spatial resolutions, to ensure the same transit time as the microwave

SM data. The MODIS products were downloaded from NASA's Land Processes Distributed Active Archive Center (LPDAAC) from the United States Geological Survey (USGS) (https://lpdaac.usgs.gov/). To be consistent with the SM data, all data were averaged by day and night products, and outliers were eliminated by the first-order difference method. Furthermore, null values were interpolated using the Savitzky-Golay filter.

3.3 Meteorological and Auxiliary Data

SM data from the China National Meteorological Station (CNMS) and China's agrometeorological and ecological observation network (http://data.cma.cn/, last access: 16 November 2019) were used to verify the downscaling SM products. We used the hourly in situ soil moisture data measured at 0-5 cm depth to investigate the accuracy of the satellite-derived surface SM estimates. Monthly products were obtained from the 2420 agrometeorological stations (including the Key-station/National Climate Observatory, Basic-station/National Meteorological Observatory and

General-station/regional meteorological station). Based on the nearest neighbor data during the daily satellite transit, and aggregated into monthly products through average to match the satellite downscaling soil moisture products, see Figure A1 for site space location. Take the AMSR series satellites used in this research as an example. The daily transit time of the satellites in China is 13:30 and 1:30. Therefore, the ground soil moisture measurements at the daytime (13:00, 14:00) and the nighttime (1:00, 2:00) are averaged. In the aggregation calculation, abnormal and unrepresentative data are

eliminated to ensure that the selected data can reflect all the physical conditions that affect the remote sensing signal. The China Ecosystem Research Network (CERN) are in different regions of the study area (Figure 1) and represent different surface and climatic conditions, which were used to validate downscaled SM.

In addition to the above data, the Land Precesses Distributed Active Archive Center (LP DAAC) of the USGS (https://lpdaac.usgs.gov/) provides digital elevation model (DEM) data with a resolution of 1 km. These data were used to





obtain terrain factors (e.g., elevation and slope) for the downscaling studies. Table 1 describes an overview of the main

data sets and a description of the corresponding variables for each data set in this study. According to the seasonal

division of weather, spring ranges from March to May, summer from June to August, autumn from September to

November, and winter from December to February.

Table 1: Overview of the data sets used in this study.

| Data sets | Satellite | Spatial/temporal resolution | Dates | Description |
|---|---|---|---|---|
| AMSR-E L3 | Aqua | 0.25°/1 month | 2002/01-2011/10 | SM |
| SMOS-IC | SMOS | 0.25°/1 month | 2011/07-2012/06 | SM |
| AMSR2 L3 | GCOM-W1 | 0.25°/1 month | 2012/07-2018/12 | SM |
| MOD11C3 | Aqua | 0.05°/1 month | 2002/07-2012/12 | LST |
| MOD13C2 | Aqua | 0.05°/1 month | 2002/07-2018/12 | NDVI |
| SRTM | - | 0.01° | - | DEM, slope |
| Station | - | 1 d | 2002/01/01-2018/12/31 | SM |

3.4 Methodology

3.4.1 Calibration and Restoration of the Satellite-derived SM

Microwave frequency and overpass time of the satellite are two important factors for deriving SM values (Cashion, et al.,

2005). In theory, the surface SM data retrieved from different frequencies have different soil sampling depths (Njoku, et al.

2003). Because the diurnal variation in SM content and temperature may be considerable, the overpass time of the sensor

can influence the retrieved SM. AMSR-E and AMSR2 have the same ascending/descending overpass times, i.e., 1:30 p.m.

and 1:30 a.m. local time. The SMOS SM retrievals occur at dawn and nightfall, corresponding to the SMOS

ascending/descending overpass times at 6:00 a.m. and 6:00 p.m. Differences in the overpass time and observed depth of

the sensors could be serious issues when matching data, particularly when deriving long-term trends. Hence, the impact of

differences among sensors is considered in this study. Despite soil moisture retrieved from AMSR-E/2 and SMOS having

different absolute values, they show similar seasonal patterns, which provides the possibility for calibrating and rescaling

to yield a long-term dataset. AMSR-E was selected as the reference in this study because it is associated with a relatively

long time series. The monthly SM average is calculated from the use of more than one and half months. The linear

regression matching technique was chosen as the calibrating method. Similar matching approaches have been successfully

used in the past (e.g., Crow and Zhan, 2007). Crow and Zhan (2007) rescaled satellite SM observations with a model by

linear regression matching, and Brocca (2013) also established regression relationships between satellites and in situ

observations for calibration of satellite observations of SM using regression matching. In general, the linear rescaling

method is realized by considering the most general linear relationship between the reference dataset (X) and the original





dataset (Y). In this study, the linear regression method was applied cell by cell and its form is Eqs. (1):

$$Y^* = \mu_X - (Y + \mu_Y)C_Y \tag{1}$$

where $\mu_X$ and $\mu_Y$ are the average values of X and Y to calculate the sequence respectively, $Y^*$ is the scaled value of the

original data Y, $C_Y$ is a scalar scaling factor, and in this study, we eliminate the largest-smallest impact in the fitting

process. Here is a linear method proposed by Yilmaz and Crow (2013) to determine the size of $C_Y$. The $C_Y$ was

calculated via Eqs. (2)

$$C_Y = \rho_{XY}\sigma_X\sigma_Y \tag{2}$$

where $\rho_{XY}$ is the correlation coefficient of X and Y, and $\sigma_X, \sigma_Y$ are the standard errors of X and Y respectively.

The calibration procedure was applied to monthly averages of SMOS-IC data. The SMOS-IC values are plotted against

the AMSR-E values for the overlapping period (07/2002 to 10/2011) to calculate calibrating parameters (linear equations).

Second, the calibrating equations derived from the previous step are applied to SMOS-IC data for the period from 2011

through 2012, producing calibrated SMOS data (SMOS reg refers to the calibrated values). The AMSR-2 values are

calibrated against the SMOS-IC values using data from the overlapping period. The equations derived from the previous

step are used to calibrate the AMSR-2 data from the period 2012 through 2018, producing AMSR-2 reg. The AMSR-E,

SMOS reg, and AMSR-2 reg data are thus obtained from 2002 to 2018.

3.4.2 Downscaling method of SM

Based on the identification of a negative correlation between the SM products and LST/NDVI, we construct a relatively

simple and efficient downscaled process, in which the TVDI is a weighting factor for downscaling. First, we computed

the fault and null value areas based on the Savitzky-Golay filter to eliminate the effects of clouds and water vapor on the

MODIS LST/NDVI images. Then, we built an LST terrain correction model to reduce the influence of terrain fluctuations

on the surface temperature inversion results. In addition, we established a monthly TVDI distribution using the

LST/NDVI inversion model based on LST and NDVI images acquired from MODIS with 0.05° spatial resolution. Finally,

we constructed an SWD model to decompose the SM pixel by pixel and generate a monthly 0.05° SM gridded product.

The structure diagram of the method is given in Figure 2.



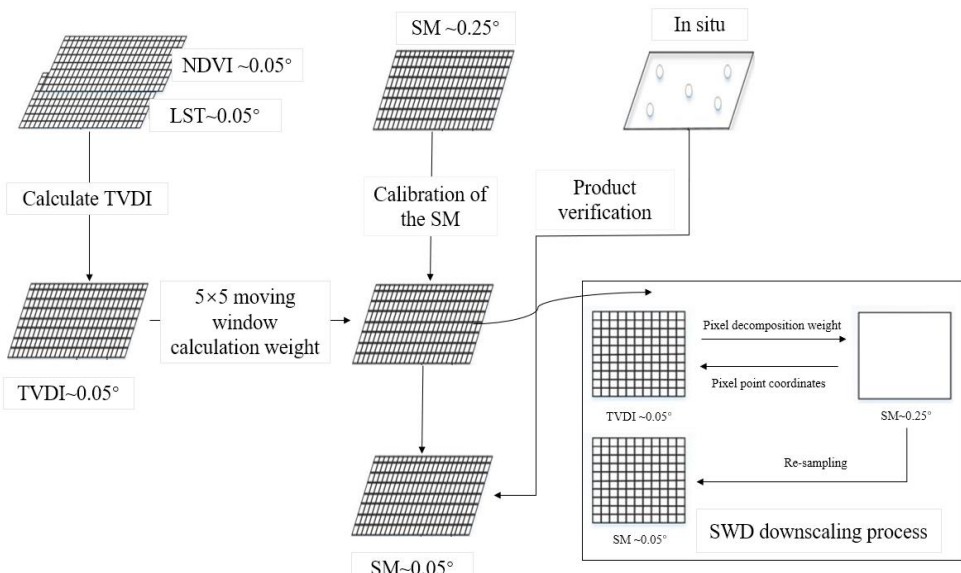

Figure 2: The flowchart for the fine SM dataset

Since optical data will be affected by clouds and harsh atmospheric conditions, there are missing and discontinuous

LST and NDVI data. To compensate for the error caused by insufficient MODIS data, the first-order difference method is

used to eliminate outliers, and then the Savitzky-Golay (S-G) filter is used to reconstruct the time series data from 2002 to

2018, and to interpolate the null values of the missing data. The specific method is shown in Eqs. (3):

$$Y_j^* = \sum_{-m}^{m} \frac{C_i * Y_{j+1}}{N} \qquad (3)$$

where $Y_j^*$ represents the time series data after the supplementation; $Y_{j+1}^*$ is equal to half the size of the smoothing

window; $C_i$ is the fitting coefficient of the Savitzky-Golay polynomial filter, i.e., the weight of the *i-th* value from the

filter head; and $N$ is the length of the data processed by the filter (the number of data points contained in the sliding

window).

Due to the large elevation variations in China, the influence of terrain on temperature must be corrected before the TVDI

can be calculated. To reduce the influence of terrain fluctuations on the surface temperature data, Eqs. (4) is used to

correct the LST products acquired from MODIS, as described in previous studies (Molero et al., 2016):

$$T_m = T_o + h \times \lambda \qquad (4)$$

where $T_m$ is the corrected surface temperature, $T_o$ is the surface temperature before correction, *h* is the elevation value

at a certain pixel, and $\lambda$ is the average influence coefficient of the elevation on the surface temperature inversion process

(where the best value of $\lambda$ is 0.006 °C/m).

    The TVDI calculation formula, which was proposed by Sandholt (2002), can adequately estimate the surface water



conditions of soil. Thus, the TVDI has been widely used in drought monitoring, and the TVDI expression is shown in Eqs.

(5), (6) and (7):


$$TVDI = \frac{T_s - T_{s\,min}}{T_{s\,max} - T_{s\,min}}$$

(5)

$$T_{s\,max} = a_1 + b_1 * NDVI$$

(6)

$$T_{s\,min} = a_2 + b_2 * NDVI$$

(7)

where $T_s$ is the LST (°C) in the study area, $T_{smin}$ is the LST of the wet side, ($a_2$, $b_2$) is the simulation coefficient of the

"wet edge" model, $T_{smax}$ is the surface temperature of the dry side, and ($a_1$, $b_1$) is the simulation coefficient of the "dry

edge" model.

Based on the LST/NDVI feature space, many studies have shown that the TVDI exhibits a significant negative

correlation with SM (Wang, 2016). The high-resolution TVDI distribution is used to weight the low-resolution SM data

pixel by pixel; then, the weight is used to decompose the low spatial resolution SM product into 0.05°SM products. The

SWD Eqs. (8) is as follows:


$$SM_i = SM_j \times \frac{1 - TVDI_a}{1 - TVDI_b}$$

(8)

where $SM_i$ represents the $SM$ data used to generate the 0.05° pixels, $SM_j$ represents the input low-resolution SM data,

$TVDI_a$ is the average TVDI value of the MODIS pixels corresponding to the SM in pixel $a$, and $TVDI_b$ is the TVDI

average of the MODIS pixels corresponding to the SM in pixel $b$.

3.4.3 Evaluation metrics of downscaled SM

It is necessary to evaluate the SM downscaling results before further application. The accuracy of the fine spatial

resolution SM is evaluated in terms of $R$, root mean square error (RMSE), bias and unbiased RMSE (ubRMSE) (Ma et al.,

2019). For this purpose, the Taylor's diagram is used to statistically summarize the correlation coefficient, centered RMSE

($E$), and normalized standard deviation (SDV) of the simulation results from the site observations by a single point in two

dimensions (2-D). The error metrics used in the study are defined as follows:


$$R = \frac{1}{N-1} \sum_{i=1}^{N} \left(\frac{T_i - T}{\sigma_T}\right)\left(\frac{L_i - L}{\sigma_L}\right)$$

(9)

$$Bias = \frac{1}{N} \sum_{i=1}^{N} (T_i - L_i)$$

(10)

$$RMSE = \sqrt{\frac{1}{N} \sum_{i=1}^{N} (T_i - L_i)^2}$$

(11)

$$ubRMSE = \sqrt{RMSE^2 - Bias^2}$$

(12)

$$E^2 = SDV^2 + 1 - 2SDV \times R$$

(13)

Correspondingly, $E$ can also be defined as:



$$E = \sqrt{\frac{ubRMSE^2}{\sigma_L}}$$

(14)

where

$$SDV = \frac{\sigma_T}{\sigma_L}$$

(15)

where $T_i$ is the downscaled SM value in the *i-th* year, $L_i$ is the in situ SM value in the *i-th* year; $T$ and $L$ are the mean

downscaled and in-situ SM values, respectively; $N$ represents the total number of observations; and $\sigma_T$ and $\sigma_L$ represent

the standard deviations of the downscaled and in-situ SM values, respectively.

In the Taylor diagram, the SDV and $R$ are expressed as the radial distance and the angle in the polar plot, respectively.

Therefore, $E$ represents the distance of the point "observed" on the Taylor diagram. The shorter the distance, the better the

consistency between the tested LST product and the reference LST observation. Only significant correlations were

considered in this study (p-value < 0.05).

## 4 Results

### 4.1 Verification of the downscaled soil moisture datasets

Figure A2 shows the soil moisture images before and after downscaling in June 2002 and the value of a cross-sectional

pixel. The results of downscaled SM products using spatial weights better retain the spatial distribution of the original

images. Specifically, the spatial details of the soil moisture data after downscaling are more delicate. It is very important

to validate the SM product before application. Because monthly products were produced in this study, we verified these

products on monthly, seasonal and annual scales. Based on a comparison of the remote sensing data with the ground

agricultural meteorological stations data, the validation reveals that the downscaled SM products boast a high accuracy at

the monthly, seasonal and annual scales (Figure 3). At these three temporal scales, the values of the ground-measured SM

were all slightly higher than those of the downscaled SM. However, due to the high variability in the SM at the seasonal

scale, measurements at the monthly and seasonal temporal scales did not show the temporal SM trends or the yearly

averaged SM values. When compared with the ground measurements, the downscaled SM data underestimated the ground

observations at all three temporal scales, especially the monthly scale. We calculated the correlation coefficients ($R$)

between the in situ and remotely sensed SM values; the $R$ value was particularly high at the yearly scale. Specifically, at

the monthly, seasonal and annual scales, the $R$ values were approximately 0.82, 0.88, and 0.90 respectively; the

corresponding ubRMSE were 0.051, 0.048 and 0.042 m³/m³ respectively. Additionally, the bias value was -0.016 m³/m³

on the annual scale, while the monthly and seasonal biases were -0.024 and -0.030 m³/m³, respectively.

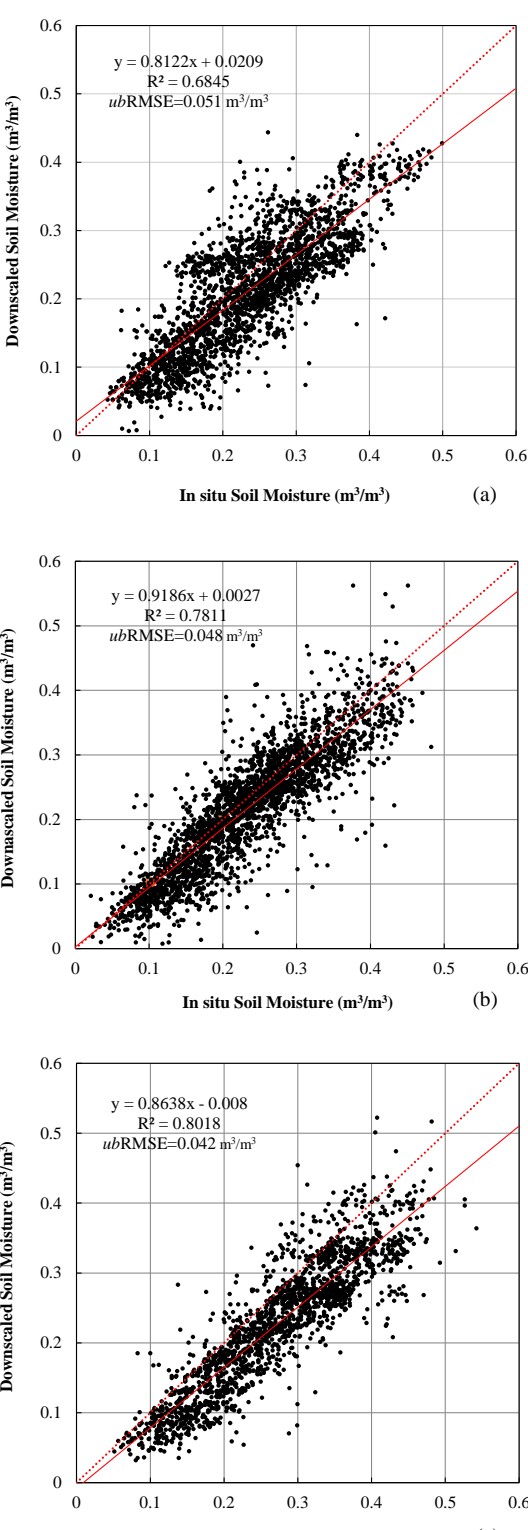


Figure 3: Correlations between the downscaled SM and in situ SM measurements at the (a) monthly, (b) seasonal and (c) annual scales.

The solid lines are the trend lines, and the dashed lines are the y=x reference lines.

Note: These comparisons do not include site data used in the correction.

The ubRMSE, bias and $R$ results for the SM products in different areas were calculated using in situ SM data. In Figure 4,

the box plots present the median of each indicator (the horizontal line within each box) and the first (Q1) and third

quantiles (represented by the bottom and top of the box, respectively). The downscaled SM is strongly correlated with the

in situ measurements, with $R > 0.52$ at most times during the 12-month period. Specifically, the downscaled SM products

had the lowest $R$ and the highest RMSE in December (possibly attributable to ice and snow cover in winter). The

downscaled SM products displayed the best correlation with in situ measurements in September (weaker vegetation

impact). Compared to the values of the North China Monsoon and Northeast Monsoon Regions, the deviation values of

the South China Monsoon and the Qinghai-Tibet Plateau Regions are more variable. The reason for this variability is not

the same: the Qinghai-Tibet Plateau some regions is covered by snow and ice around the year, while South China features

dense surface water networks and abundant rain. Please note that, the soil moisture data of frozen soil region is somewhat

questionable. In order to maintain the integrity of the data and previous study demonstrated that the JAXA AMSR-E/2

products still have some capability to capture the temporal tend of soil moisture in frozen seasons (Zeng et al., 2015), we

keep it. Therefore, the follow-up verification and analysis process also follow this criterion.

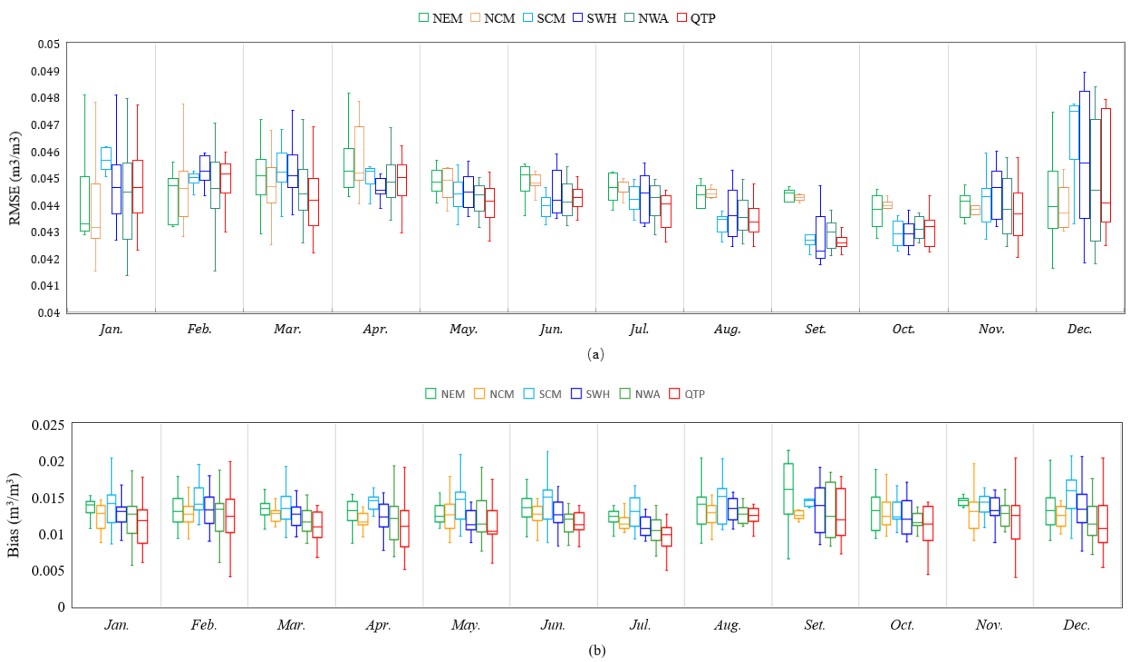



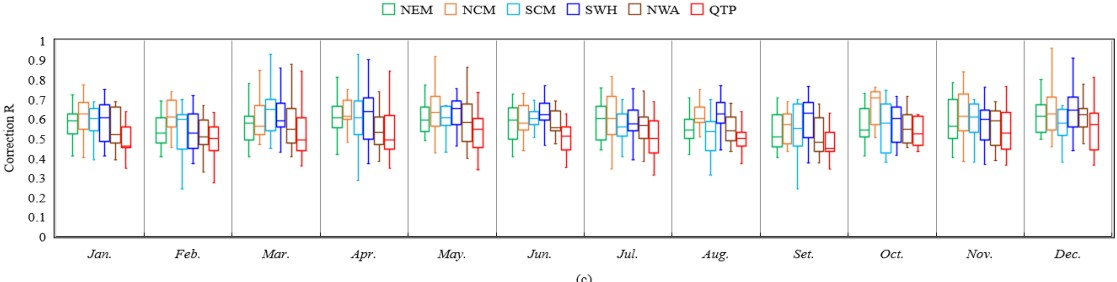

Figure 4: Box plots of the RMSE, bias and *R* (p<0.05) of comparison between downscaled SM and in situ SM in each region.

As a complement to the evaluation presented in the previous section, the performance of the downscaled SM product was evaluated against in situ surface SM observations from stations in different land cover areas. The performance of the downscaled SM at the selected stations is provided, and the performance criteria (including *R*, SDV and center RMSE) estimated between the downscaled SM and in situ observations are also reported in Taylor diagrams in Figure 5. In

general, the dots (stations) are unevenly distributed in the Taylor diagrams for all regions, indicating that the downscaled SM accuracy varies from one station to another. Forest land estimates more frequently plot outside of one normalized SDV circle than other estimates, indicating that SM associated with higher vegetation is more variable.

In eastern China (the Northeast Monsoon, North China Monsoon, and South China Monsoon Regions), the downscaled SM products are in good agreement with the ground observations, although the variability of a few stations is large. Most

of the correlation (*R*) values between the downscaled SM and the in situ observation range between 0.6 and 0.9. In western China, i.e., the Qinghai-Tibet Plateau and the Northwest Arid Regions, the downscaled products have poor correlations with the in situ observations, but higher correlation values were generally obtained in low vegetation areas.



Figure 5: Taylor diagrams illustrating a statistical comparison between downscaled SM and in situ measurements in each region.



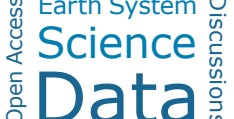

**4.2 Spatiotemporal change of SM in different natural regions of China**

Over the past 17 years, the national average SM content was approximately $0.093 m^3/m^3$ and exhibited an overall decreasing trend ($b = -0.167$, $R = 0.750$, $P = 0.05$). This result also explains the increase in temperature, which resulted in an increase in evaporation and thus a decrease in SM in the context of global warming. From 2002 to 2012, there were slight fluctuations, but after 2013, the SM content sharply decreased. In addition, the years with the highest and lowest

SM contents were 2004 (11.07%) and 2016 (7.31%), respectively. From the annually averaged SM content of each subregion over the past 17 years, the SM values in the South China Monsoon Region were much higher than those in the other regions (average of 16.46%). In this region, the values of 2002-2011 were consistent with the national average, and the values of 2011-2013 were higher than the national average. The significant decline in SM in this region shows that it was more affected by the monsoon than the other regions in China. In contrast, the average annual trends of the North

China Monsoon Region and Northeast China Monsoon Region, which are also affected by the monsoon, were relatively stable. The Southwest Wet Region ranked second, with an average SM of 9.16%, followed by the Northeast Monsoon Region and the North China Monsoon Region, with average SM values of 8.69% and 8.44%, respectively. Furthermore, the Northwest Arid Region and the Qinghai-Tibet Plateau Region displayed consistently low SM averages of 6.87% and 6.34%, respectively. The accuracy of the SM products in the Qinghai-Tibet Plateau Region was relatively low, mainly due

to the greater impact of snow cover. This analysis shows that the SM contents of monsoon-affected areas (i.e., the Northeast China Monsoon Region, North China Monsoon Region and South China Monsoon Region) are more sensitive than those of inland areas (Northwest Arid Region and Southwest Wet Region). The average SM content was highest in the South China Monsoon Region, which also showed the most significant change. This region displayed a decreasing trend throughout the study period. The rate of decline in this region is defined by ($b= -0.246$, $R=0.570$, $P=0.01$) and is

much higher than the rates in the other monsoon regions. The North China Monsoon Region has experienced numerous droughts in the past and is currently exhibiting a decreasing trend ($b=-0.383$, $R =0.621$, $P=0.05$). Thus, it is predicted that the drought in North China will further intensify and even trigger a series of agricultural disasters. The SM contents in the Southwest Wet Region and the Northeast China Monsoon Region have slightly decreased over the past 17 years, but the Northwest Arid Region has shown a slight increasing trend ($b=0.04$, $R=0.651$, $P=0.05$). The drought situation in the

northwestern part of the study area has positive significance for ecological, agricultural and livestock production in the Northwest Arid Region of China.

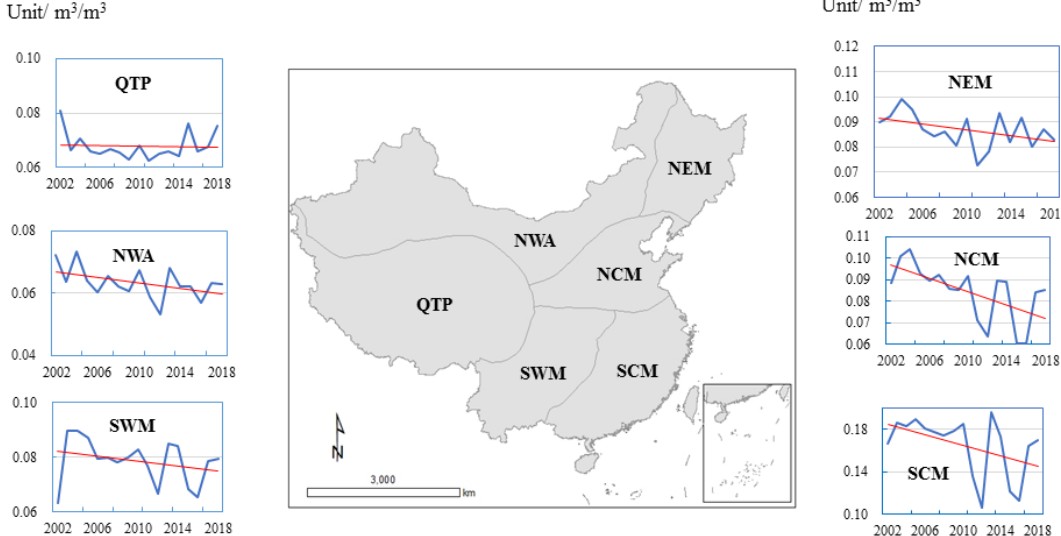

Figure 6: Interannual variations in the average annual SM of the six geographic regions

Note: Due to the lack of data from January to June of 2002, the annual averages are calculated from the average of only the second half; the same procedure applies to the analyses below.


### 4.3 Characteristics of the spatiotemporal variations in SM

To obtain the overall spatial long-term annual, seasonal and monthly variations of downscaled surface SM in detail, a linear regression was conducted at the pixel level from 2002 to 2018. The slope is used to represent the variation rate of the downscaled SM. Figure 7 (left) is a distribution map of the correlation coefficients, and Figure 7 (right) shows the

changes in slope corresponding to the correlation coefficients. The SM changes in China have exhibited obvious geographical and seasonal differences over the past 17 years. Different slopes indicate different trends. Specifically, a slope >0 indicates an increasing trend, with a higher value indicating a more pronounced change; a slope <0 indicates a decreasing trend, with a lower value indicating a more pronounced change, and a slope=0 indicates no change. Based on the annual SM content, the overall SM content in China has shown a decreasing trend, with the area of significant

reduction accounting for 45.9% of the total area and the area of significant increase accounting for 17.5% of the total area. From the perspective of the different regions, relatively obvious decreasing trends are present in the plain areas west of the Changbai Mountains in the Northeast China Monsoon Region, the Liaodong Peninsula and Shandong Peninsula in the North China Monsoon Region, the eastern coastal areas and the middle and lower reaches of the Yangtze River Basin and the Sichuan Basin in the Southwest Wet Region, and the forest areas in the southern Qinghai-Tibet Plateau Region, and

the slopes of these changes exceed 0.3 ($R<-0.6$). Similar findings have also been reported in previous studies (Mao, et al., 2008a; Liang, et al., 2017). This phenomenon occurred mainly because during this period, Southwest China experienced high temperatures and extensive evaporation, and these conditions contributed greatly to a regional water deficit for plants. In contrast, the SM contents in the southern Hexi Corridor, the southern part of Xinjiang and the northern part of the Qinghai-Tibet Plateau in the Northwest Arid Region increased significantly, with slope values of 0.2, which is less than



$R$>0.5. From 2002 to 2018, the SM contents in most parts of China showed a decreasing trend (except for the Northwest

Arid Region), which is consistent with the analysis in Figure 6. This result shows that drought risk will increase

throughout the country in the future. In the Northwest Arid Region, the conditions will become more humid, which will

alleviate the current drought situation in northwestern China (Cong, et al., 2017). Furthermore, effectively improving the

ecological environment in the Northwest Arid Region of China has positive significance for western China's development

and the Belt and Road Initiative.

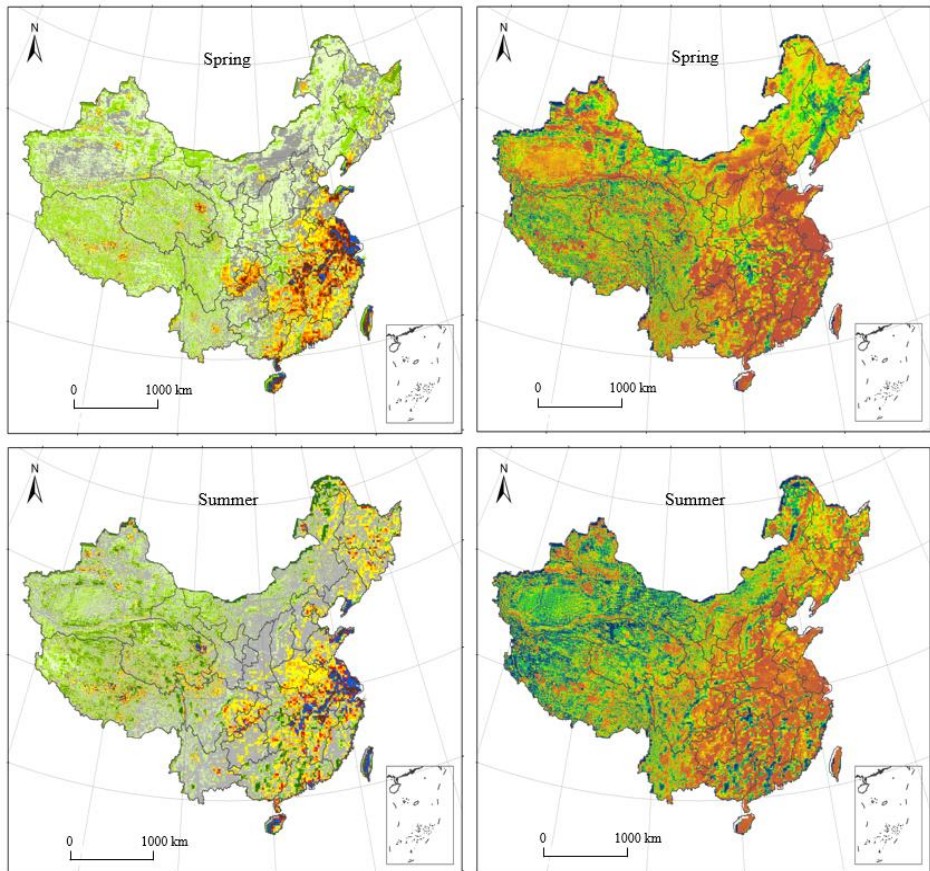

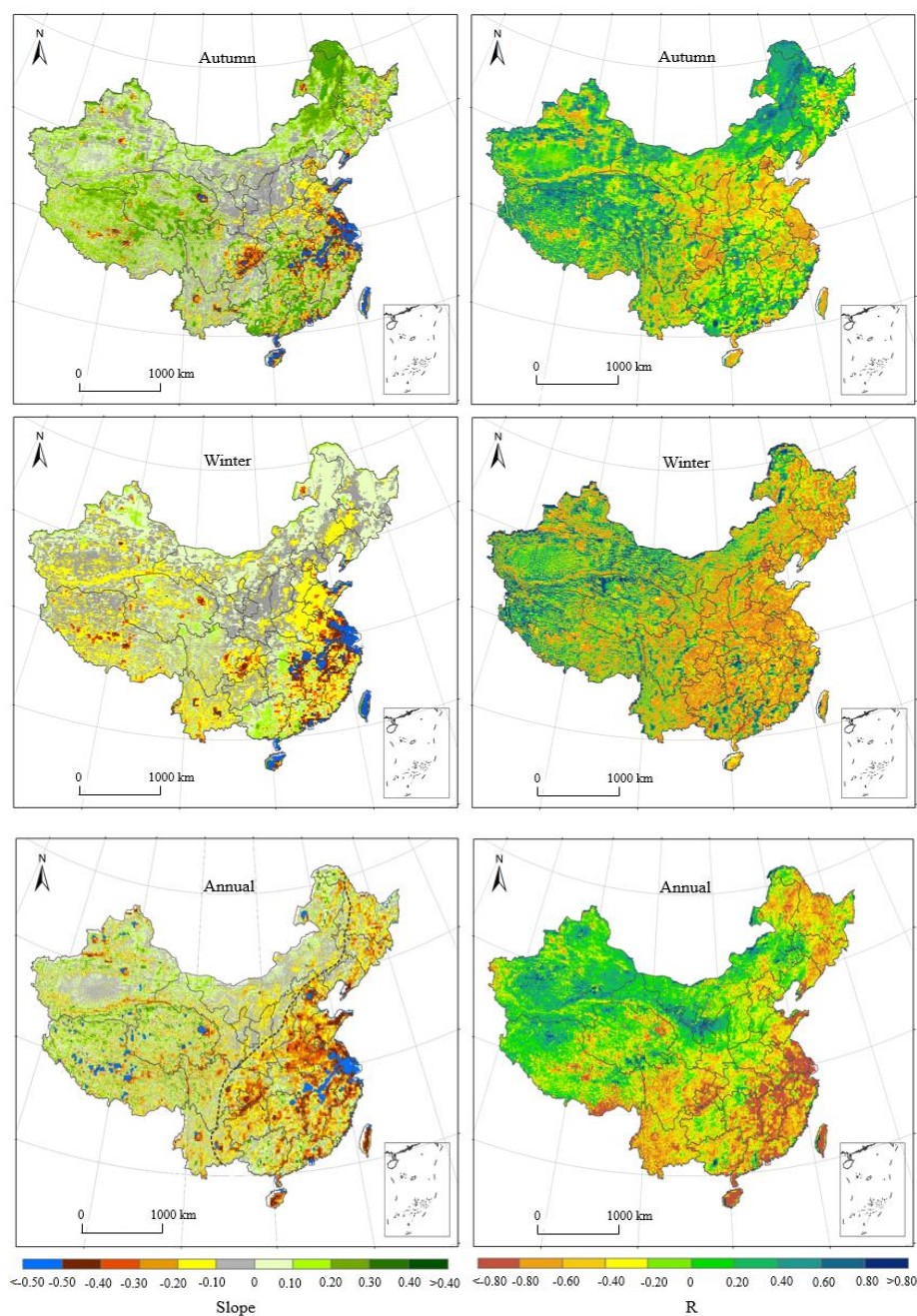

Figure 7: The interannual variability rates (slope, left) and correlation coefficients (*R*, right) of the seasonal and annually averaged SM

430            contents from 2002 to 2018; the dashed lines divide areas of high and low slope values.

To better understand the SM changes throughout China, the spatial distributions of the annual variations in the SM for

different regions in different seasons were analyzed. As shown in Figure 8, the overall changes in the SM throughout

China over the past 17 years exhibited obvious seasonal characteristics. In spring, the rate of the decrease in SM was

relatively high, except in some areas of the Northeast Plain, the Three Gorges Reservoir area, and the area surrounding the





Central Gobi region. More than 80% of the South China Monsoon Region showed different decreasing trends, especially

in the Jiang-Huai region, and the value of the slope reached as high as -0.3 (*R*<-0.7). Another severe decline occurred in

the Sichuan Basin during spring. In addition, in the Liaodong Peninsula, the Shandong Peninsula, and the eastern part of

the Kunlun Mountains in the North China Monsoon Region, a decreasing trend predominated with a negative slope of less

than -0.2 (*R*<-0.5). In summer, the heterogeneity of the SM among the six subregions was obvious: large change trends

occurred in the northwestern Heilongjiang region of the Northwest Arid Region, the Yunnan-Guizhou Plateau in the

Southwest Wet Region, and the Yangtze River plain in the South China Monsoon Region; the slope was less than -0.4

(*R*<-0.7). Generally, due to the large changes in the interannual hydrothermal and monsoon precipitation, the area affected

by the monsoon in the east varied greatly. From spring to summer, the range of fluctuation in the SM content in the South

China Monsoon Region was significantly enhanced. Usually, shifts in precipitation belts occur during the rainy season;

these shifts are governed by the summer monsoon and occur during the rainy season in the Pearl River Delta and Yangtze

River Delta (Zhou et al., 2010). During the rainy season, the total rainfall was approximately 80% of the annual rainfall

(Yan et al., 2015). The SM contents in the Heilongjiang region in the Northeast Monsoon Region in autumn showed an

extreme downward trend with a slope of less than -0.4 (*R*< -0.7). This trend extended to the northern part of Inner

Mongolia in the Northwest Arid Region (slope<-0.4, *R*<-0.5), and a significant downward trend was also apparent in the

hilly area of Chongqing, which is on the border between the Southwestern Wet Region and the South China Monsoon

Region, with values that decreased to less than -0.3 (*R* < -0.5). In addition, in the Yellow-Huai River area of the North

China and South China Monsoon Regions, the SM in the middle and lower reaches of the Yangtze River and the Sichuan

Basin in the Southwest Wet Region showed a significant decreasing trend, and the slope in the main area decreased to less

than -0.4 (*R*<-0.5). In summer and autumn, the trends in the monsoon regions were obvious. Although many rainfall

events occur in the summer and autumn monsoon regions, the spatial and temporal distributions of precipitation were not

balanced. In addition, the middle and lower reaches of the Yangtze River are mainly dominated by a subtropical

high-pressure system in summer, during which a large amount of evaporation takes place, which may have been the main

cause of the observed decline. The change in SM in winter was not as significant as that in other seasons. The decline

occurred mainly in Southwest China (such as Yunnan and western Guangxi), as was also detected in previous studies

(Mao, et al. 2012). The precipitation in autumn was generally low, except in the vegetation areas in the south, and there

was a rising slope (slope>0.4, *R*>0.7) in the Three Gorges Dam in the upper reaches of the Yangtze River. The appearance

of a decreasing slope (slope<-0.4, *R*<-0.7) is very noteworthy: it is very likely that the Three Gorges Dam will have a

significant impact on the local SM variations after storage. The slopes during the four seasons in the Bohai Rim region of

the North China Monsoon Region and the Yangtze River Delta region in the South China Monsoon Region were

extremely high, which may have been caused by the rapid increase in the area of impervious surfaces attributable to

extensive urbanization. Conversely, the SM content increased in areas affected by monsoons, such as the Qinghai-Tibet

Plateau Region (South Tibet) and the Northwest Arid Region (East Inner Mongolia).



We performed an analysis of the spatiotemporal changes in SM on the monthly scale in different years (Figure 8). The monthly average variability was more volatile than the variability at the seasonal and annual scales, especially in January

and July. In January, northern China is under the influence of the Eurasian high-pressure center, and the monsoon in July is under the control of the subtropical Pacific high-pressure system. The South China Monsoon Region is particularly susceptible to extreme weather events (e.g., El Niño occurred in 2006 and 2015), resulting in weak summer monsoons and southerly monsoon rains in central China or south of the Yangtze River. In the northern regions, droughts and high temperatures are prone to occur in summer, and low temperatures and floods are prone to occur in the south. The low and

high temperatures and heavy precipitation caused by these climatic conditions may also have been important causes of the sudden changes in SM.

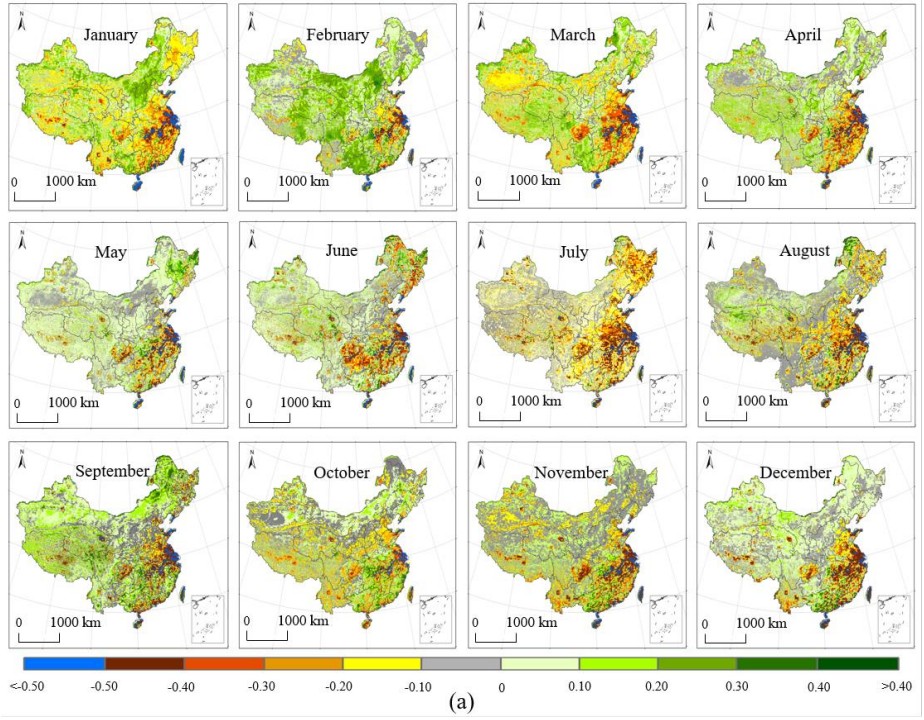

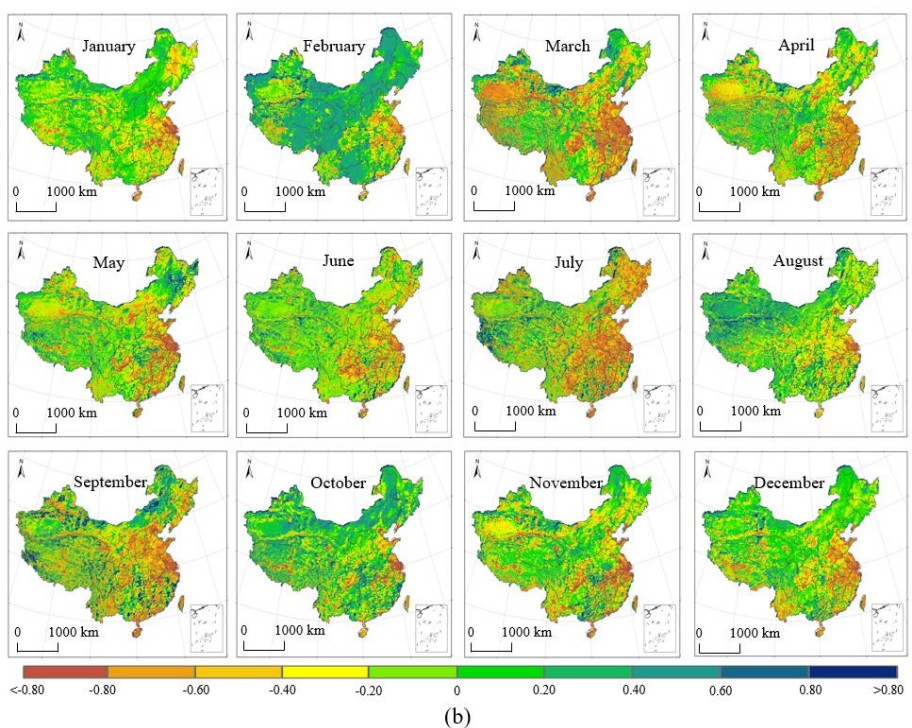

Figure 8: The interannual variability rates (slopes) and correlation coefficients (R) of the monthly average SM contents from 2002 to 2018.
480         (Note: Due to the lack of data for the spring seasons of 2002 and 2018, these two phases of data are not included in the calculations.)

## 5 Data availability

The fine-resolution SSM dataset presented in this article Creative Commons Attribution 4.0 International is available at the following link:

http://doi.org/10.5281/zenodo.4049958 (Meng et al., 2020). It covers all of China land area at a monthly temporal resolution and a 0.05º

spatial resolution from 2002 to 2018.

## 6 Discussion and conclusions

Global climate change has modified the spatial and temporal distributions of China's hydrologic resources, which in turn

have led to changes in the Earth's biochemical cycle. SM is an important component of the Earth's biochemical cycle and

not only is an important driver of the global water cycle but also potentially affects global atmospheric circulation. Access

to quantitative SM information enhances not only water management, agricultural productivity, and drought monitoring

capabilities but also climate prediction. Studying the high-resolution spatial and temporal characteristics of SM is of great

significance for practical applications, such as water resource management, agricultural estimations, drought monitoring and

climate change. Based on the 25-km spatial resolutions of the AMSR-E, SMOS, and AMSR2 microwave products, an SWD

model was established using the negative correlation between the SM and TVDI for spatial data fusion to generate

0.05°-resolution long-sequence continuous SM products. The accuracy of the verified data set is satisfactory (bias: -0.024,

-0.030 and -0.016 m3/m3, ubRMSE: 0.051, 0.048 and 0.042, correlation coefficient (R): 0.82, 0.88, and 0.90 on monthly,

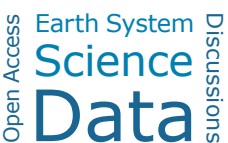

seasonal and annual scales, respectively). The data were used for a comprehensive spatial and temporal analysis of the SM status, which revealed the characteristics and differentiation of SM in China's natural regions from 2002 to 2018.

Downscaled products were used to analyze the spatial and temporal differences in SM among the six natural regions in China over the past 17 years, the results of which indicated that the SM changes in China have obvious regional and seasonal characteristics.

SM showed an overall decreasing trend, and there were some fluctuations in SM in China over the past 17 years; these fluctuations can be divided into a slow growth phase from 2002 to 2011 and a strong declining phase from 2011 to 2013.

From 2014 to 2018, the SM steadily increased, and a slow decreasing phase occurred in 2010. These findings mean that SM is currently slightly decreasing, and in the next few years, China will face the risk of increased drought (especially during summer in the Southeast Monsoon Region and the North China Monsoon Region). Rapid decreasing trends occurred in the North China Monsoon Region, the South China Monsoon Region, the Yangtze River Delta region and the Bohai Sea region, while significant increasing trends occurred in the southern part of the Northwest Arid Region (in the northwestern

Qinghai-Tibet Plateau). These trends can be summarized as wet in the south and dry in the north, with increases in the west and decreases in the east. In the different seasons, although the overall trends were still declining, the SM changed significantly from spring to winter. The SM was relatively evenly distributed throughout the six subregions in spring, while the SM decreased in the eastern monsoon region (the Northeast Monsoon Region, North China Monsoon Region, and South China Monsoon Region) in summer and autumn. Moreover, the inland areas (some areas in the Northwest Arid Region, the

Qinghai-Tibet Plateau Region, and the Southwest Wet Region) showed an opposite trend, indicating the significant impact of summer monsoon precipitation on SM. In autumn, the SM was significantly reduced in the northeastern part of China, and grasslands dried out. Overgrazing and grassland reclamation exacerbated desert conditions. These conditions may have led to the rapid decline in SM. The monthly average change was basically the same as the seasonal average variation, but the changes were more severe.

Increasing urbanization has a significant impact on SM, especially in areas with relatively rapid urbanization, such as the middle and lower reaches of the Yangtze River, the Pearl River Delta and the Bohai Rim, but not areas influenced by the monsoon. One of the factors that cannot be ignored is the decline in SM caused by the rapid expansion in the area of impervious surfaces caused by rapid urbanization. The increasing precipitation and artificial afforestation in the Northwest Arid Region of China have led to an increase in SM throughout this region. Surface temperatures can not only affect the

evapotranspiration of SM but also indirectly affect SM by affecting the transpiration of vegetation. Therefore, temperature also has a significant impact on the changes in SM. All these analyses indicate that it is very important to analyze the spatiotemporal characteristics of SM for local climate change research.

**Supplement.** The supplemental material related to this article is available online at:

**Author contributions.** KM and FM designed the research and developed the methodology; KM and XM supervised the downloading and

processing of satellite images; XM wrote the manuscript; and JS, JZ, XS, YC, LJ, ZG and all other authors revised the manuscript.



**Competing interests.** The authors declare that they have no conflicts of interest.

**Acknowledgments.** The authors thank the China Meteorological Administration for providing the ground measurements, the Japan Aerospace Exploration Agency for providing the AMSR-E and AMSR2 SM data, the ESA's Earth Observation User Services Portal of the European Aviation Administration for providing the SMOS-IC SM data, the NASA's Earth Observing System Data and Information System for providing the MODIS LST and NDVI data and DEM data. This work was supported by the National Key Project of China (Nos. 2018YFC1506602, 2018YFC1506502), Fundamental Research Funds for Central Non-profit Scientific Institution (Grant No. 1610132020014), and Open Fund of State Key Laboratory of Remote Sensing Science (Grant No. OFSLRSS201910).

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
