# Peer review of "A fine-resolution soil moisture dataset for China in 2002-2018"

_Earth System Science Data, 2020_

## Short Comment (SC1) · 9 Jan 2021

In this paper, authors presented a new method for producing long-term surface soil moisture data in China, and the data is well-validated. However, I am a bit confused about the interannual trend of soil moisture in this study. According to this dataset, surface soil moisture declined in most parts of China during 2002~2018, especially in eastern China (Figure 6 and Figure 7). However, we noticed that GPM IMERG precipitation increased in China, especially in eastern China (please see the figure we attached below) where your data however, indicated a sharp decline in surface soil moisture. Considering the strong positive impact of precipitation on surface soil moisture, this conflict seems quite strange. In addition, we also noticed a recently published global long-term microwave-based surface soil moisture dataset called RSSSM

(https://doi.org/10.5194/essd-13-1-2021). That dataset seems to suggest an increase in surface soil moisture in China over the same period, while the eastern China showed an obvious increasing trend, which generally agrees with the spatial pattern of the trend of precipitation. So, I may doubt which dataset is more reliable in terms of the interannual trend. Theoretically, the 'linear regression matching technique' applied in your study can harmonize the absolute values, or the spatial patterns of different soil moisture products, but may be less capable of calibrating and harmonizing the interannual variations of different products retrieved from different sensors. The interannual trends of different microwave soil moisture products may differ a lot, probably because the disturbances of major influence factors (e.g., vegetation, open water) on different soil moisture retrievals are quite different, in fact.
Fig. 1.

---

## Author Comment (AC1) · 28 Jan 2021

In this paper, authors presented a new method for producing long-term surface soil moisture data in China, and the data is well-validated. However, I am a bit confused about the interannual trend of soil moisture in this study. According to this dataset, surface soil moisture declined in most parts of China during 2002-2018, especially in eastern China (Figure 6 and Figure 7). However, we noticed that GPM IMERG precipitation increased in China, especially in eastern China (please see the figure we attached below) where your data however, indicated a sharp decline in surface soil moisture. Considering the strong positive impact of precipitation on surface soil moisture, this conflict seems quite strange. In addition, we also noticed a recently published global long-term microwave-based surface soil moisture dataset called RSSSM (https://doi.org/10.5194/essd-13-1-2021). That dataset seems to suggest an increase in surface soil moisture in China over the same period, while the eastern China showed an obvious increasing trend, which generally agrees with the spatial pattern of the trend of precipitation. So, I may doubt which dataset is more reliable in terms of the interannual trend. Theoretically, the "linear regression matching technique" applied in your study can harmonize the absolute values, or the spatial patterns of different soil moisture products, but may be less capable of calibrating and harmonizing the interannual variations of different products retrieved from different sensors. The interannual trends of different microwave soil moisture products may differ a lot, probably because the disturbances of major influence factors (e.g., vegetation, open water) on different soil moisture retrievals are quite different, in fact.

**Response: Thank you very much for your good comments and suggestions. We did a comprehensive analysis, and improved data quality. We have tried to modify our manuscript and improve the quality of paper.**

   **Our work in this manuscript is mainly to do downscaling research and provide a set of higher resolution (0.05°) soil moisture products based on the soil moisture products of Japan Aerospace Exploration Agency (JAXA) which has been verified. Because the resolution of passive microwave scale is too low, the theoretical model for large scale (mixed pixels) is not very mature. Although**

there are many soil moisture algorithms and products, and different algorithms have their own advantages and disadvantages, and their accuracy performance is inconsistent in different regions. As you mentioned, especially in areas with a lot of vegetation and rainfall, the accuracy performance is inconsistent.

Usually in vegetation coverage areas, single albedo and optical thickness coefficient values are obtained differently for different retrieval algorithms, which result in a relatively large difference in soil moisture retrieval.

Another difference is the treatment of heavy rainfall. When there is heavy rainfall, the retrieval error of microwave soil moisture is very large. Some retrieval algorithms determine that when there is heavy rainfall, the retrieval soil moisture is an invalid value (usually set to a null value), but some algorithms directly set the soil moisture saturation value as the soil moisture value.

Due to the above two reasons, especially the second reason, there will be deviations in the calculation of the monthly average and the annual average, because invalid values are not included in the calculation for algorithm. In other words, when there is a heavy rainfall, the invalid value in the soil moisture product is not included in the calculation.

We have made an analysis. The soil moisture products you mentioned (RSSSM https://doi.org/10.5194/essd-13-1-2021) has some advantages in the monthly and annual calculation of soil moisture in the eastern rainfall area of China, and it is more reasonable to handle during heavy rainfall. The resolution of this data set is 0.1°. We optimize the value of the high-precision area of this data set to our previous data set, and then further downscaling. In this way, we can have a set of soil moisture data sets with higher accuracy and resolution. Thank you very much.

---

## Short Comment (SC2) · 21 Feb 2021

Our National Satellite Meteorological Center is responsible for the production of FY satellite passive microwave soil moisture products. Thank you for producing this set of high-resolution and long-term soil moisture products in China. By comparing with some product data and site data, the accuracy of this product has been significantly improved in some local areas. For soil moisture products, although satellite soil moisture remote sensing inversion has made great progress, we need to let users understand the specific limitations of current satellite products. I hope the author can emphasize these. It is very important to study soil moisture changes for weather forecasting and agricultural production. In particular, high-resolution soil moisture product data is key parameter for evapotranspiration and water conservation studies on small and medium-scale wa-

tersheds. There are currently three main methods to obtain soil moisture product data at the regional scale in China. The first is through thermal infrared and visible light re-mote sensing data, namely the Drought Index (TVDI). Although this product has a high resolution and can show changes in soil moisture on a small scale, it is not very com-parable in different regions, which is mainly due to the limited penetration of thermal infrared. The second is radar (Active microwave) remote sensing. Active microwave has some advantages, but it is greatly affected by roughness. The third is passive microwave soil moisture retrieval product. Generally speaking, the accuracy of pas-sive microwave retrieval of soil moisture is higher than that of active microwave, which is currently recognized as the best method for inversion of soil moisture in large re-gions. Now the mature passive microwave products in China mainly include AMSR-E, AMSR2, SMOS, FY, etc. Different frequency settings of microwave sensor and different incident angles will result in large differences in the inverted soil moisture information. The L-band has a stronger penetrating power and represents a deeper soil moisture depth. The frequency and incident angle of AMSR-E and AMSR2 sensors are basically the same, their similarity soil moisture information is the highest, which keep them have a relatively high consistency. Although the frequency setting of FY is basically similar to AMSR-E and AMSR2, the incident angle is slightly different. In addition, the low frequency band of FY passive microwave sensor is not very stable, and the inversion error is relatively large especially in high vegetation coverage regions. SMOS mainly uses the L-band with different incident angles to invert soil moisture. Because the fre-quency and observation angle are very different from the previous three, the inverted soil moisture is quite different from the previous three instruments, which represents a deeper depth of soil moisture information. In addition, the soil moisture algorithms developed by different sensors are different, and different algorithms obtain some pa-rameters, especially the vegetation coverage regions, which will cause differences in soil moisture products. For microwave sensors mounted on polar-orbiting satellites, the soil moisture products retrieved are all instantaneous products, and the satellite passes twice a day. Therefore, the transit time of different satellites is inconsistent,

and the weather conditions are also inconsistent, resulting in inconsistent soil moisture products. Therefore, it is more difficult to produce a long-term soil moisture product with high precision with high spatiotemporal resolution, and we should consider consistency of time, observation angle and observation depth. Although this study considered theses, it did not elaborate. The author is recommended to emphasize the following three points. 1. What are the differences in the characteristics of sensors mounted on different satellites, including frequency and incident angle. 2. What are the advantages and disadvantages of different sensor soil moisture products, the depth of soil moisture that they represent, and how to calibrate the depth information of different sensor. 3. In the discussion part of this paper, add an explanation of the limitations of this data set. In general, this data set selects the relatively stable passive microwave soil moisture data products, and uses thermal infrared and visible light data to further downscale and make further corrections. This has greatly improved the resolution of passive microwave products in China and provided key parameters for many studies, especially agricultural drought monitoring and analysis.

---

## Author Comment (AC2) · 8 Mar 2021

Thank you very much for your good comments and suggestions. This research was under the support of China's national key research and development program "Global Meteorological Satellite Remote Sensing Dynamic Monitoring, Analysis Technology and Quantitative Application Methods and Platform Research" and "Multi-source Meteorological Data Fusion Technology Research and Product Development". The purpose is to provide a set of soil moisture data sets with high spatial and temporal resolution. The development of the data set considered a variety of factors and overcome many difficulties. There are some differences in the soil moisture retrieved by different sensors, mainly due to the band settings and observation angles of different satellite instruments. The lower the frequency, the deeper the depth of the observed soil

moisture information. In addition, different inversion algorithms have different considerations of land surface temperature and vegetation and rainfall, which will also lead to inconsistent inversion results. I have reproduced the data set and the new version will be uploaded soon. The revised manuscript also highlighted your suggestions. Considering that the instrument frequency, incident angle, and observation time are as consistent as possible, we try to absorb the advantages of different coarse-resolution soil moisture data sets, perform downscaling inversion, and perform verification and spatio-temporal analysis. Many users, especially in China, have given high evaluations to the high-resolution downscaling data set we provide. Thanks again.

---

## Author Comment (AC3) · 13 Mar 2021

We have absorbed the advantages of the RSSSM dataset (https://essd.copernicus.org/articles/13/1/2021/), (https://doi.pangaea.de/10.1594/PANGAEA.912597), further optimized and downscaled, and generated a new high-temporal-resolution dataset version 2 (https://zenodo.org/record/4588293#.YEweeTmd2Ul), along with the previous data set, which has been downloaded 1262 times and received good comments from some users. The key R&D project team of China has conducted applications and tests and believes that the accuracy of the new data set is better. Thank you for your valuable suggestions, which greatly improved the quality of our data sets and manuscript.

---

## Author Comment (AC4) · 15 Mar 2021

This data set was completed with the support of China's 13th Five-Year Key R&D Plan, which has been downloaded 1481 times (https://zenodo.org/record/4049958#.YE6hljmd0dU), and we have received many user's suggestions and good comments. A new high-temporal-resolution dataset version 2 (https://zenodo.org/record/4588293#.YEweeTmd2Ul) has been reproduced and provided to users, and we have also made corresponding changes to manuscript. The key R&D project team of China has conducted applications and tests and believes that the accuracy of the new data set is better Thanks to everyone for valuable suggestions, which greatly improved the quality of our data sets and manuscript.

Thanks again.

---

## Referee Comment (RC1) · Anonymous Referee #1 · 6 Apr 2021

Soil moisture is an important indicator for a wide range of applications, especially those related to agriculture and the ecosystem. Measuring soil moisture is expensive with ground stations and difficult to cover the spatial variation of soil moisture. Satellite observation is able to cover a large area and has the potential for deriving soil moisture for a country or even the globe; however, the coarse resolution of the satellite-derived datasets limits wide use of these data, especially for soil moisture derived from passive microwave observations. The paper presents a soil moisture dataset for China with monthly and 0.05 degree resolutions, which could be helpful to many applications.

The paper claims that the 0.05-degree spatial resolution is a breakthrough for soil moisture; however, datasets with the same resolutions have been published for even the

globe, for example:

Chen, Yongzhe, Xiaoming Feng, and Bojie Fu. "An Improved Global Remote-Sensing-Based Surface Soil Moisture (RSSSM) Dataset Covering 2003–2018." Earth System Science Data 13, no. 1 (January 5, 2021): 1–31. https://doi.org/10.5194/essd-13-1-2021.

Jing, Wenlong, Pengyan Zhang, and Xiaodan Zhao. "Reconstructing Monthly ECV Global Soil Moisture with an Improved Spatial Resolution." Water Resources Management 32, no. 7 (May 2018): 2523–37. https://doi.org/10.1007/s11269-018-1944-2.

The paper needs to compare to the existing dataset to investigate the accuracy and improvements of this dataset over others, to provide a subjective review of the dataset accuracy for users. Meanwhile, recent studies have suggested advantage of downscaling soil moisture using machine learning methods, for example:

Liu, Yangxiaoyue, Wenlong Jing, Qi Wang, and Xiaolin Xia. "Generating High-Resolution Daily Soil Moisture by Using Spatial Downscaling Techniques: A Comparison of Six Machine Learning Algorithms." Advances in Water Resources 141 (July 2020): 103601. https://doi.org/10.1016/j.advwatres.2020.103601.

The dataset was produced using aged algorithms with little methodology innovation by the authors, making it difficult to justify the value of this manuscript.

Many of the sentences make little sense, and the writing needs improvement before publication.

It appears that the authors have been misusing terms in the paper, such as "verify". Please replace it with "validation" or "evaluation".

The authors interpreted the changes of derived soil moisture in China, and explain the changes due to climate changes. I wonder if the authors have compared the results to climate datasets to justify these statements?

Line 80, please add a citation at the end of the sentence.

Line 89, replace "verified" with "validated".

Line 103, why "most of the areas in China"? What happened to the rest of China?

Line 136, replace "finite working life" with "limited lifespan".

Line 225, please clarify "different absolute values".

Line 227-228. Please clarify how the average was calculated from the one and half month's data.

Line 378, what "consistent" is referring to? Similar trend or similar values?

Line 386, I am not sure what "sensitive" refers to.

Line 388, "most significant change" please clarify.

Line 520, is there evidence to support the statement regarding the effects of urbanization?

---

## Author Comment (AC5) · 10 Apr 2021

Dear Reviewers and Editors,

Thank you for your valuable comments on our manuscript. First, we would like to express our sincere appreciation for your professional and insightful remarks on our paper.

The global soil moisture dataset is constantly being produced, especially in recent years, the frequency of updates is getting faster and faster. Each soil moisture dataset and method of producing soil moisture has its own advantages and disadvantages. Our soil moisture dataset is mainly concentrated in China. Two similar sensors mounted on

different satellites are used to produce a set of soil moisture datasets that are continuous in time and space in China. For the missing part in the middle, a relatively reliable sensor was used to make up for it. In order to ensure the consistency of the time and depth of the observation data of the three instruments, we have made corrections through building reconstruction model. In particular, we took advantage of the site data obtained from the National Meteorological Administration to make local improvements. In order to meet the needs of research such as agricultural drought monitoring, we downscaled the soil moisture products and obtained a higher resolution dataset.

This dataset was completed with the support of China's 13th Five-Year Key R&D Plan, and one of the goals of this project is to produce a set of soil moisture datasets with high spatial and temporal resolution in China, which has been downloaded 1667 times (https://zenodo.org/record/4049958#.YE6hljmd0dU). Validation and application analysis show that the accuracy of our dataset is relatively high, and we have received good comments from many users.

Your comments are all valuable and have helped us to improve the quality of our paper. We have studied each comment and have made revisions that we hope will meet with approval. Please find our detailed responses in attachment. Revisions to the manuscript will be highlighted in blue in the revised manuscript file. Thanks again.

With our best regards,

Xiangjin Meng and co-authors

Please also note the supplement to this comment:
https://essd.copernicus.org/preprints/essd-2020-292/essd-2020-292-AC5-supplement.pdf

**Supplement:**

Dear Reviewers and Editors,

Thank you for your valuable comments on our manuscript. First, we would like to express our sincere appreciation for your professional and insightful remarks on our paper.

The global soil moisture dataset is constantly being produced, especially in recent years, the frequency of updates is getting faster and faster. Each soil moisture dataset and method of producing soil moisture has its own advantages and disadvantages. Our soil moisture dataset is mainly concentrated in China. Two similar sensors mounted on different satellites are used to produce a set of soil moisture datasets that are continuous in time and space in China. For the missing part in the middle, a relatively reliable sensor was used to make up for it. In order to ensure the consistency of the time and depth of the observation data of the three instruments, we have made corrections through building reconstruction model. In particular, we took advantage of the site data obtained from the National Meteorological Administration to make local improvements. In order to meet the needs of research such as agricultural drought monitoring, we downscaled the soil moisture products and obtained a higher resolution dataset.

This dataset was completed with the support of China's 13th Five-Year Key R&D Plan, and one of the goals of this project is to produce a set of soil moisture datasets with high spatial and temporal resolution in China, which has been downloaded 1667 times (https://zenodo.org/record/4049958#.YE6hljmd0dU). Validation and application analysis show that the accuracy of our dataset is relatively high, and we have received good comments from many users.

Your comments are all valuable and have helped us to improve the quality of our paper. We have studied each comment and have made revisions that we hope will meet with approval. Please find our detailed responses below. Revisions to the manuscript will be highlighted in blue in the revised manuscript file. Thanks again.

With our best regards,
Xiangjin Meng and co-authors

###########################################################################

**Response to referees**

**Response to referee #1**
   # Summary:

> Soil moisture is an important indicator for a wide range of applications, especially those related to agriculture and the ecosystem. Measuring soil moisture is expensive with ground stations and difficult to cover the spatial variation of soil moisture. Satellite observation is able to cover a large area and has the potential for deriving soil moisture for a country or even the globe; however, the coarse resolution of the satellite-derived datasets limits wide use of these data, especially

for soil moisture derived from passive microwave observations. The paper presents a soil moisture dataset for China with monthly and 0.05 degree resolutions, which could be helpful to many applications.

**Response:** We would like to thank you for reviewing our manuscript. Your comments and good suggestions are very important for us to improve the quality of manuscript and dataset. We have carefully addressed all the issues raised by you and the response is presented below. As you mentioned, soil moisture products are very important, so many countries and regions (especially the United States, the European Union, Japan, China, etc.) have been developing soil moisture products and are constantly updating their versions, especially in recent years, the frequency of updates is getting faster and faster. Traditional methods and new methods have been promoting the development of this product. Various methods have their own advantages and disadvantages, and the accuracy of different products is not consistent in different places.

In vegetation coverage areas, especially forest coverage areas, the accuracy of different algorithms is somewhat different. The inconsistent consideration of vegetation in traditional physical algorithms leads to inconsistent soil moisture retrieval, while machine learning methods (such as deep learning algorithms) are difficult to obtain the corresponding large-scale ground truth values and the accuracy is also difficult to guarantee, so many people have been thinking of ways to improve retrieval accuracy of vegetation coverage area.

1. The paper claims that the 0.05-degree spatial resolution is a breakthrough for soil moisture; however, datasets with the same resolutions have been published for even the globe, for example:
(1) Chen, Yongzhe, Xiaoming Feng, and Bojie Fu. "An Improved Global Remote-Sensing-Based Surface Soil Moisture (RSSSM) Dataset Covering 2003–2018." Earth SystemScience Data 13, no. 1 (January 5, 2021): 1–31. https://doi.org/10.5194/essd-13-1-2021.
(2) Jing, Wenlong, Pengyan Zhang, and Xiaodan Zhao. "Reconstructing Monthly ECVGlobal Soil Moisture with an Improved Spatial Resolution." Water Resources Manage-ment 32, no. 7 (May 2018): 2523–37. https://doi.org/10.1007/s11269-018-1944-2.
(3) Zhang, Q., Yuan, Q., Li, J., Wang, Y., Sun, F., and Zhang, L.: Generating seamless global daily AMSR2 soil moisture (SGD-SM) long-term products for the years 2013–2019, Earth Syst. Sci. Data, 13, 1385–1401, https://doi.org/10.5194/essd-13-1385-2021, 2021.
(4) Kang, C.S., Zhao, T., Shi, J., Cosh, M.H., Chen, Y., Starks, P.J., Collins, C.H., Wu, S., Sun, R. and Zheng, J., 2020. Global Soil Moisture Retrievals From the Chinese FY-3D Microwave Radiation Imager. IEEE Transactions on Geoscience and Remote Sensing. https://doi.org/10.1109/TGRS.2020.3019408

**Response: Thank you for your guidance. Although many people have done a**

**lot of excellent work, the focus of different work is inconsistent and the work we did is different from other dataset in terms of time scale and spatial resolution, especially our focus on improving product accuracy in China with the aid of a large number of ground station data, and the resolution is about 0.05-degree resolution covering 2002–2018. The resolution and time length of other datasets are introduced as follows.**

(1) Chen et al. (2021) developed the global remote-sensing-based surface soil moisture dataset (RSSSM) **covering 2003–2018** through a neural network approach, and the accuracy is very high and **resolution is about 0.1-degree resolution**, **which has not been downscaled. This dataset can be publicly downloaded.**

(2) Jing et al. (2018) proposed a two-steps reconstruction approach for reconstructing satellite-based soil moisture products (ECV) at an improved spatial **0.05-degree resolution covering 2001–2012**. The reconstruction model implemented the Random Forests (RF) regression algorithm to simulate the relationships between soil moisture and environmental variables, and takes advantages of the high spatial resolution of optical remote sensing products, which is a good work. **This dataset cannot be publicly downloaded.**

**(3)** Zhang et al. (2021) develop a novel spatio-temporal partial convolutional neural network (CNN) for AMSR2 soil moisture product gap-filling, and the resolution is **about 0.25-degree resolution covering 2013–2019. This dataset can be publicly downloaded.**

(4) The research group of my supervisor Shi Jiancheng who is one of co-authors has improved the algorithm for FY 3D microwave data and also produced a global soil moisture product (Kang et al. 2020), and **the resolution is about 0.25-degree resolution covering 20017–2019. This dataset cannot be publicly downloaded.**

These works are very good, and do not contradict the work we have done, and their respective focuses are different. Our main point is to emphasize the consistency and downscaling of soil moisture obtained from three similar sensors mounted on different satellites, as well as their application in China. Different microwave bands have different penetration capabilities for different vegetation, and it is still limited to obtain soil moisture information under dense vegetation cover. Therefore, it is still difficult to guarantee the inversion accuracy of either traditional physical methods or machine learning methods in some regions. We mainly want to use ground observation sites to further calibrate the soil moisture products in vegetation areas, especially forest-covered areas, to improve the accuracy of soil moisture products in China. We have introduced and revised our manuscript.

2. The paper needs to compare to the existing dataset to investigate the accuracy and improvements of this dataset over others, to provide a subjective review of the dataset accuracy for users. Meanwhile, recent studies have suggested

advantage of down-scaling soil moisture using machine learning methods, for example:

Liu, Yangxiaoyue, Wenlong Jing, Qi Wang, and Xiaolin Xia."Generating High-Resolution Daily Soil Moisture by Using Spatial Downscaling Techniques: A Comparison of Six Machine Learning Algorithms." Advances in Water Resources 141 (July2020): 103601. https://doi.org/10.1016/j.advwatres.2020.103601.

Response: Thank you for your good suggestions. Some data is available for download, and some data is not available for download. We have mentioned it in the previous question. Our dataset differs from other data sets in observation time, resolution, and length of the data set, and we have tried made revisions and comparison.

Liu et al. (2020) have made a good comparison analysis which is a good work. Each method has its own advantages and disadvantages, and sometimes the accuracy of different algorithms in different areas is inconsistent. There are two main reasons. One is the data reason, and the other is the algorithm reason. At present, the resolution of most passive microwave data is very low, and they are basically mixed pixels. Most algorithms basically do not consider the problem of mixed pixels. Especially in vegetation coverage areas, it is difficult for us to completely eliminate the influence of vegetation and obtain high-precision ground soil moisture, and machine learning algorithms are no exception which is highly dependent on the accuracy of the training dataset. In areas covered by vegetation, we still need to further improve the accuracy of soil moisture products. We try our best to do more supplementary analysis on our data products. Thanks again.

3. The dataset was produced using aged algorithms with little methodology innovation by the authors, making it difficult to justify the value of this manuscript. Many of the sentences make little sense, and the writing needs improvement before publication.

Response: Thank you for your guidance, and we have tried our best to modify it. The main purpose of our work is to use AMSR-E, AMSR2, and SMOS soil moisture products to construct a long-term sequence of soil moisture products, eliminating the difference in observation time and depth of different products. In order to further meet the needs of agricultural drought and other research, we further downscaled and improved the data resolution by constructing a weight decomposition model using thermal infrared and visible light bands, which can ensure the stability of soil moisture product. In addition, in vegetation areas, especially forest coverage areas, we used site data to make corrections. Machine learning methods have certain advantages, but sometimes they are not stable enough. For example, when there is the influence of clouds and rainfall. Neural network is not a new method which has existed decades ago and has long been used in geophysical parameter inversion. Neural networks have their own advantages, but the disadvantage is that they depend on the accuracy of the training dataset. For

large-scale pixels, it is difficult to obtain the corresponding ground truth values. The same is true for large-scale validation. In many cases, it is relative comparison.

We believe that by making the dataset public, and then application make it public in the paper, so that many users can use it in practice and point out that there are problems for data, which can help greatly improve the accuracy of dataset. Thanks gain.

4. It appears that the authors have been misusing terms in the paper, such as "verify". Please replace it with "validation" or "evaluation".

Response: Thank you for your guidance. We have made revisions.

5. The authors interpreted the changes of derived soil moisture in China, and explain the changes due to climate changes. I wonder if the authors have compared the results to climate datasets to justify these statements?

Response: Thanks for your guidance. Most of our analysis is similar to many related researches on climate change in China, but a few of them are inconsistent with some studies. Using different data sources and different research perspectives, it is normal to get a small number of different conclusions. This requires us to compare and analyze the reasons through different studies and even different disciplines in the future, so as to promote scientific progress. Controversy and differences are good things, which can make more people think about the reasons for the differences through public papers and data in ESSD, and thus discover more interesting knowledge. Most geophysical products (including NASA products) have been constantly updated and improved. We have been working hard to improve data accuracy, and we will continue to update the dataset version through user application feedback.

6. Line 80, please add a citation at the end of the sentence.

Response: Thank you for your guidance. We have made revisions.

7. Line 89, replace "verified" with "validated".

Response: Thank you for your guidance. We have made revisions.

8. Line 103, why "most of the areas in China"? What happened to the rest of China?

Response: Thank you for your guidance. We have made revisions.

9. Line 136, replace "finite working life" with "limited lifespan".

Response: Thank you for your guidance. We have made revisions.

10. Line 225, please clarify "different absolute values".

Response: Thank you for your guidance. We have made revisions.

11. Line 227-228. Please clarify how the average was calculated from the one and half month's data.

Response: Thank you for your guidance. We have made revisions.

12. Line 378, what "consistent" is referring to? Similar trend or similar values?

Response: Thank you for your guidance. We have made revisions.

13. Line 386, I am not sure what "sensitive" refers to.

Response: Thank you for your guidance. We have made revisions.

14. Line 388, "most significant change" please clarify.

Response: Thank you for your guidance. We have made revisions.

15. Line 520, is there evidence to support the statement regarding the effects of urbanization?

Response: Thank you for your guidance. We have made revisions.

---

## Referee Comment (RC2) · Anonymous Referee #2 · 22 Apr 2021

Soil moisture is very crucial in exploring the response of vegetation dynamics to the climate change in ecohydrological processes. Having professed my general enthusiasm for the topic and its importance, I have some concerns that require substantive effort, a large part should be recalculated and rewrote. The paper should be suitable for publication following the recommend major revisions below: 1. There are several similar studies on the reference. What are new findings on this study? 2. Line 111-114: The authors should explain more about how to divide into these six regions (NEM, NCM, SCM, etc.) by climate and topography, and add these in Fig 1. What does the different color mean is Fig 1 (the patch not the points). 3. Eqs4: Why do you need to correct the surface temperature? Did these data already include the elevation effects? 4. Eqs8: These letters (i, j, a and b) seem to be four different pixels. Please explain clearly about

how to get the new soil moisture data from the previous soil moisture and TVDI in different location. 5. Line 327-328: Can you explain why for this: 'the downscaled SM data underestimated the ground observations at all three temporal scales, especially the monthly scale.' 6. Lin 343: 'possibly attributable to ice and snow cover in winter' If it is true, why the RMSE in April was higher than that in November. 7. There are five elements including station, vegetation types, R, SDV and RMSE in Fig 5. It is difficult for the reader to get information easily and it did not show the highlight (SM associated with higher vegetation is more variable) well. 8. Line 378-379: Would it be affected by the soil types either? The soil types could also affect the soil moisture. Please include the effect of soil types in this part. 9. Fig6: I found large interannual variability along with the declined trend in each panel. Please explain the variability in the main text. Is it caused by low quality of the data or climate change? 10. Please mark the 'Changbai Mountains', 'Liaodong Peninsula', 'Sichuan Basin', etc. in relevant figures. 11. Fig8: why did the soil moisture increase in Northeast China in autumn? 12. The authors actually did 'results and discussion' in 'results' session. The 'Discussion and conclusions' part did not show much discussion. Please modify the session name.

---

## Author Comment (AC6) · 5 May 2021

Soil moisture is very crucial in exploring the response of vegetation dynamics to the climate change in ecohydrological processes. Having professed my general enthusiasm for the topic and its importance, I have some concerns that require substantive effort, a large part should be recalculated and rewrote.

Response: Thank you for your valuable comments on our manuscript, which are very important and helped us to improve the quality of dataset and paper. **We have upgraded the dataset based on the comments of reviewers and many users' feedback in the process of applying data, and it is currently updated to version 3 and has been downloaded 1716 times (https://zenodo.org/record/4738556#.YJJcs8DiuUk). We have received good comments from many users,** and studied each comment and have made revisions. Please find our detailed responses below. Revisions to the manuscript will be highlighted in blue in the revised manuscript file. Thanks again.

Best regards,
Xiangjin Meng and co-authors

The paper should be suitable for publication following the recommend major revisions below:

1. There are several similar studies on the reference. What are new findings on this study?

**Response:** In addition to providing a set of soil moisture data sets with high spatiotemporal resolution for agricultural drought monitoring and climate change models, we also analyzed the spatiotemporal changes in different regions. In the past 17 years, China's soil moisture has shown cyclical fluctuations and a downward trend, but the northwest has a slightly wet trend. Global warming drives the intensification of the water cycle, which is the fundamental reason for the warming and humidification of the climate in Northwest China. For the Northwest, water vapor mainly comes from the Arabian Sea and the Indian Ocean. As the Arctic warms, water vapor from the Arctic Ocean increases. Under the action of air currents, water vapor in the three places concentrated in the northwest, and precipitation in the northwest increased rapidly, which leads to an increase in soil moisture. For more specific analysis, please refer to the manuscript.

2. Line 111-114: The authors should explain more about how to divide into these six regions (NEM, NCM, SCM, etc.) by climate and topography, and add these in Fig 1. What does the different color mean is Fig 1 (the patch not the points).

**Response:** We mainly divide regions based on topography (elevation), rainfall and other factors, so that we can better evaluate soil moisture changes and make corrections and assessments based on corresponding ground observation sites. Different color means different elevation information.

3. Eqs4: Why do you need to correct the surface temperature? Did these data already include the elevation effects?

**Response:** There are a lot of clouds in some mountainous areas in the south, and a few places have very few effective values of surface temperature. Only the bottom of the mountain has an observation station. In order to obtain a more reliable temperature at high altitudes, we can only use the valley ground observation site temperature, so the influence of elevation must be considered in the correction process.

4. Eqs8: These letters (i, j, a and b) seem to be four different pixels. Please explain clearly about how to get the new soil moisture data from the previous soil moisture and TVDI in different location.

**Response:** Thank you for your guidance. We have made revisions. Revisions to the manuscript will be highlighted in blue in the revised manuscript file.

5. Line 327-328: Can you explain why for this: 'the downscaled SM data underestimated the ground observations at all three temporal scales, especially the monthly scale.'

**Response:** Compared with thermal infrared remote sensing, passive microwave remote sensing has less influence on clouds and rainfall. Especially when there is rainfall, the accuracy error of passive microwave inversion of soil moisture is still relatively large. Therefore, when it is judged that there is rainfall and the error is relatively large, the inversion algorithm usually sets this inversion value to an invalid value (0). Some algorithms are filled by neighboring pixels. In addition, there is a gap between the data of two adjacent scenes, and some algorithms fill it with adjacent pixels. When there is rainfall, most of the soil moisture value reaches the maximum saturation value. The processing method of the inversion algorithm will lead to underestimation of soil moisture content during rainfall, which will lead to underestimation of soil moisture at all three temporal scales, especially the monthly scale. In view of this defect, we have performed optimized calculations to improve the accuracy as much as possible.

6. Lin 343: 'possibly attributable to ice and snow cover in winter' If it is true, why the RMSE in April was higher than that in November.

**Response:** In most areas, RMSE is relatively larger in winter. However, in some places, due to more rain and clouds in the spring, the error of the satellite inversion of soil moisture is relatively large, resulting in relatively large errors. In addition, it has something to do with the site location and measurement. Due to the low resolution of passive microwaves, there is a big difference between the representativeness of the site and the real pixels, which will also cause deviations in the validation process. We have already made more analysis and explanation in the manuscript.

7. There are five elements including station, vegetation types, R, SDV and RMSE in Fig 5. It is difficult for the reader to get information easily and it did not show the highlight (SM associated with higher vegetation is more variable) well.

**Response:** Thank you for your guidance, we have made revisions and gave a more detailed description.

8. Line 378-379: Would it be affected by the soil types either? The soil types could also affect the soil moisture. Please include the effect of soil types in this part.

**Response:** Thank you for your guidance. The type of soil (surface type) has also a certain impact on the change of soil moisture, and we have made a supplementary explanation in the manuscript.

9. Fig6: I found large interannual variability along with the declined trend in each panel. Please explain the variability in the main text. Is it caused by low quality of the data or climate change?

**Response:** There may be many reasons, but the most important reason is global warming. The overall increase in temperature leads to an increase in evaporation, so that the overall soil moisture has a slight downward trend. The change trend of soil moisture is fluctuating, and the performance is different in different places. After considering the impact of rainfall, we re-corrected and re-analyzed the data in the manuscript.

10. Please mark the 'Changbai Mountains', 'Liaodong Peninsula', 'Sichuan Basin', etc. in relevant figures.

**Response:** Thank you. We have made revisions.

11. Fig8: why did the soil moisture increase in Northeast China in autumn?

**Response:** Global warming drives the intensification of the water cycle, which is the fundamental reason for the warming and humidification of the climate in Northwest China. For the Northwest, water vapor mainly comes from the Arabian Sea and the Indian Ocean. As the Arctic warms, water vapor from the Arctic Ocean increases. Under the action of air currents, water vapor in the three places concentrated in the northwest, and precipitation in the northwest increased rapidly, which leads to an increase in soil moisture.

12. The authors actually did 'results and discussion' in 'results' session. The 'Discussion and conclusions' part did not show much discussion. Please modify the session name.

**Response:** Thank you for your guidance. We have made revisions.